# Simulated contrail processed aviation soot aerosol are poor ice nucleating particles at cirrus temperatures

Baptiste Testa[1], Lukas Durdina[2], Jacinta Edebeli[2], Curdin Spirig[2], and Zamin A. Kanji[1]

[1]Institute for Atmospheric and Climate Science, ETH Zürich, Zürich, Switzerland
[2]Centre for Aviation, ZHAW School of Engineering, Winterthur, Switzerland

**Correspondence:** Baptiste Testa (baptiste.testa@env.ethz.ch) and Zamin A. Kanji (zamin.kanji@env.ethz.ch)

**Abstract.** Aviation soot surrogates processed in contrails are believed to become potent ice nuclei at cirrus temperature. This is not verified for real aviation soot, that can have vastly different physico-chemical properties. Here, we sampled soot particles from in-use commercial aircraft engines and quantified the effect of contrail processing on their ice nucleation ability at $T <$ 228 K. We show that aviation soot becomes compacted upon contrail processing but this does not change their ice nucleation ability in contrast to other soot types. The presence of $H_2SO_4$ condensed in soot pores, the highly fused nature of the soot primary particles and their arrangement limit the volume of pores generated upon contrail processing, limiting sites for ice nucleation. Furthermore, we hypothesized that contrail processed aviation soot particles emitted from alternative jet fuel would also be poor ice nucleating particles if their emission sizes remain small ($< 150$ nm).

## 1 Introduction

Aviation soot particles directly emitted in the upper troposphere at cirrus temperatures have been considered as potential ice nucleating particles (INPs) impacting cirrus cloud properties hence affecting the earth radiative budget (Lee et al., 2021). However, current aviation soot radiative forcing estimates are associated with large uncertainties, arising mainly from their unconstrained ice nucleation abilities (Righi et al., 2021). Yet, the ice nucleation properties of real aviation soot have been quantified only recently for the first time, and they suggest a high likelihood of poor ice nucleation ability at cirrus relevant temperatures ($T < 235$ K) (Testa et al., 2024). This suggests that aviation soot would not perturb the formation of background cirrus clouds, and that current radiative forcing estimates need to be updated.

Soot particles can also be re-emitted into the upper troposphere via contrails that form in the exhaust wake of aircraft (Kärcher, 2018). Once the contrail sublimates, soot residuals inside the contrail ice crystals are released with potentially different properties. Capillary forces arising from water condensed in soot pores (Ma et al., 2013), may induce the collapse of the soot aggregates (China et al., 2015; Bhandari et al., 2019; Corbin et al., 2023) promoting the formation of small aggregate voids. Soot residuals from contrails have been shown to enhance ice nucleation of aviation soot proxies (Mahrt et al., 2020b; Gao et al., 2022b) due to increased porosity upon aggregate compaction, potentially allowing them to compete with background aerosols for cirrus cloud formation.

At cirrus temperatures, soot particles nucleate ice via the 3-step pore condensation and freezing (PCF) mechanism. First, water

vapor condenses into soot pores below water saturation (relative humidity over liquid water [$RH_w$] < 100 %), followed by the homogeneous ice nucleation of the pore water, and then growth of the ice out of the pore (Marcolli, 2014; Marcolli et al., 2021; Christenson, 2013). The pore size must be small enough to trigger capillary condensation but larger than the ice germ to allow its nucleation. Pore of relevant diameters for PCF are in the range of 2-30 nm (Marcolli et al., 2021) and fall into the mesopore size range of 2-50 nm (Haul, 1982).

Depending on the soot properties (coating, primary particle arrangement), the number of PCF-relevant mesopores in aggregates might increase as the result of compaction, increasing the probability of PCF to occur (Gao and Kanji, 2022a, b; Zhang et al., 2020; Nichman et al., 2019; Mahrt et al., 2020b). Testa et al. (2024) showed that aviation soot ice nucleation properties are different from the surrogates used in past studies, thus contrail processing from the previous studies are likely not representative of aviation soot contrail processing. In this study, we quantify the ice nucleation ability of contrail processed aviation soot particles sampled from in-use commercial aircraft engines, with an emphasis on their morphological change upon processing. The effect of aviation soot mixing state on the change in morphology and ice nucleation is also investigated.

## 2 Experimental Method

### 2.1 Aviation soot sampling, processing and ice nucleation measurements

The measurements were conducted at the aircraft engine maintenance and testing facility SR Technics at the Zürich Airport. The experimental setup was designed to simulate targeted atmospheric processes and to mimic the contrail processing of the sampled aviation soot particles (Fig. 1a). Limitations of the ground setup in representing atmospheric processes (e.g., aircraft exhaust evolution, contrail formation) and impact on the soot ice nucleation ability are discussed in Sect. 5. Soot particles were sampled from in-use commercial aircraft engines (multiple models from Pratt & Whitney and CFM International) running in an indoor test cell, with air intake at ambient temperature and humidity. The engines were all fueled with Jet A-1 fuel and ran from low to high power (30-100 % sea level thrust). The detailed sampling system is described in Testa et al. (2024) which we briefly describe here. The engine exhaust gas and particles were sampled by a heat-resistant alloy probe ~1 m downstream of the engine exhaust nozzle and directed by a long trace-heated line (12 m at 433 K) to a stirred tank, that was in a room next to the engine test cell, and acted as an aerosol reservoir where the soot particles accumulated and coagulated (Fig. 1b). The exhaust temperature drops in three steps, from the engine temperature (thousands of Kelvin) to the heated line temperature (433 K) and then to room temperature followed by a third drop from room temperature to the cloud chamber temperature (< 228 K, see below). Engine emission lasted about 20 min and ice nucleation measurements several hours. The soot size distribution reached in the tank after coagulation was different for each engine owed to the different soot emission indices of the engines tested. The average mode electrical mobility diameter ($D_m$) $\pm 1\sigma$ was 250 $\pm$50 nm with minimum and maximum mode diameters of 80 and 450 nm. Contrail processing of the soot particles was conducted following an identical experimental procedure described in e.g., Mahrt et al. (2020b) and Gao and Kanji (2022a). In brief, the polydisperse aerosols were directed from the aerosol reservoir to a first cloud chamber (Lacher et al., 2017; Mahrt et al., 2018) set to contrail cloud thermodynamic conditions ($T$ = 228 K and $RH_w$ = 105 %, HINC1 in Fig. 1b) allowing the soot particles to activate and freeze to ice crystals. The sample

flow downstream of HINC1 was then directed into a subsaturated flow tube, CATZ (cloud aerosol transitioner Zurich, $T = 233$ K and $RH_i = 56$ %) allowing the ice crystals formed in HINC1 to sublimate. At this stage, the sample flow included contrail

processed soot and interstitial soot particles that may not have formed ice in HINC1. Then, contrail processed and interstitial aerosols were directed into a second cloud chamber (HINC2 in Fig. 1b) whose relative humidity (RH) was varied from ice to liquid water supersaturation. Aerosol to sheath flow in the cloud chambers was 1:10 to 1:12. The number of soot particles entering HINC2 and ice crystals detected downstream of HINC2 were counted by a scanning mobility particle sizer (SMPS, Classifier 3082, Column 3081, CPC 3776 or 3775, TSI Inc., flow mode 0.3 L min$^{-1}$, SMPS2 in Fig. 1b) and by an optical

particle counter (OPC GT-526S, MetOne), respectively. The fraction of the soot particles nucleating ice, the activated fraction (AF) is defined as:

$$AF = \frac{\#\,Ice\,crystals}{\#\,Soot\,particles} \tag{1}$$

For a part of the experiments, the soot volatile fraction ($H_2SO_4$ and organics) was denuded with a catalytic stripper (CS08, Catalytic Instruments, $T = 623$ K) prior entering HINC1. In total, four aviation soot populations were studied: "unprocessed" soot, catalytically stripped soot ("CS-soot"), contrail processed soot ("CP-soot"), and catalytically stripped plus contrail processed soot ("CS-CP-soot"). The ice nucleation ability for the unprocessed and CP-soot samples were systematically measured for all engine tested (13 engines in total). The ice nucleation ability of the CS-soot and CS-CP-soot were measured for 5 engines.


## 2.2   Characterization of aviation soot morphology

Aviation soot particle morphology was investigated with transmission electron microscopy (TEM) for few targeted engines. The particles were dried with molecular sieve driers and collected onto TEM grids (copper Formvar/Carbon, TED PELLA INC.) with a nanoparticle TEM sampler (Partector TEM, naneos particles solution GmbH). Unprocessed and processed (i.e.,

CS-, CP- and CS-CP-) soot were collected onto different TEM grids at location TEM1 in Fig. 1b for unprocessed and CS-soot and TEM2 for CP- and CS-CP-soot. The Partector TEM sampler uses a soft particle impaction technique (electrostatic precipitation), limiting the effect of the sampling process on the particle morphology. Soot aggregate breaking upon impaction was not observed on the various soot samples. Only clearly isolated soot particles on the grid were imaged to avoid imaging aggregated particles on the grid. Individual soot aggregates were imaged with a JOEL-JEM 1400 microscope and the images processed

with the MATLAB (R2020a, MathWorks Inc., Natick, USA) code from Dastanpour and Rogak (2014). The latter was modified to derive convexity, circularity, aspect ratio, maximal 2D projected aggregate length ($L$), width ($W$) and equivalent spherical diameter ($D_{2D,eq}$) as described elsewhere (Bhandari et al., 2017; China et al., 2013, 2014; Mahrt et al., 2020b; Testa et al., 2024). In addition, a SMPS (SMPS1 in Fig. 1b) monitored aerosol size distributions upstream of HINC1, allowing to compare the impact of the various processing on the particle sizes.

Particle mass of size-selected unprocessed and CP-soot particles was measured with a centrifugal particle mass analyser

(CPMA; Cambustion Ltd., Cambridge, UK) in tandem with a DMA and CPC similar to Abegglen et al. (2015) and Durdina et al. (2014). From the mass measurements, information on the morphology of the particles can be extracted. As soot particles are fractal, their mass scale with their diameter following a power law relationship (e.g., Abegglen et al., 2015, and reference therein):

$$m_{\mathrm{p}} = C\, D_{\mathrm{m}}{}^{D_{\mathrm{fm}}} \tag{2}$$

where $m_{\mathrm{p}}$ is the mass of the particles with electric mobility diameter $D_{\mathrm{m}}$, $C$ is a constant called the mass-mobility prefactor, and $D_{\mathrm{fm}}$ is the mass-mobility exponent. The latter is a parameter describing the aggregate morphology, with $D_{\mathrm{fm}} = 3$ for spherical aggregate and 1 for infinite chain-like aggregate. We note that this relation holds for soot particles with constant primary particle sizes. This is not strictly true for aviation soot particles sampled in this study as shown by the TEM images (see exemplary TEM images in Figs. A1 and A2). In addition, the relation assumes that the particles were formed in the same environment, which holds in our study since all soot populations undergo similar processes in the combustor and in the aerosol reservoir (Fig. 1b).

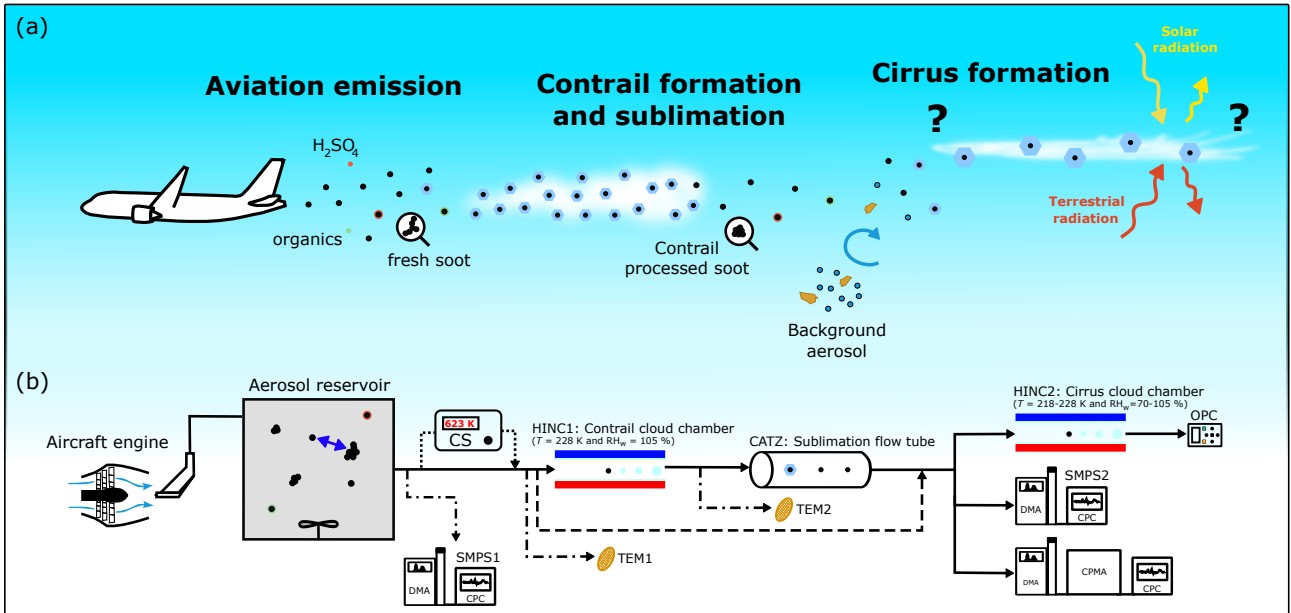

**Figure 1.** (a) Schematic of main atmospheric processes associated with aviation soot-cirrus interaction. From left to right: Due to the incomplete combustion of aviation fuel, aircraft engines emit soot particles externally and internally mixed with $SO_2/H_2SO_4$ and organics (unburned hydrocarbons, oil droplets). Plume particles (mainly soot and $H_2SO_4$) can activate as cloud droplets and ice crystals (Kärcher, 2018; Voigt et al., 2021) forming a contrail cloud due to high concentration of water vapor and low temperature which can immediately sublimates or persist and subsequently sublimate. The soot ice residuals released upon sublimation are referred to as contrail processed soot particles. Background aerosol (e.g., dust, solution droplets) and soot can both nucleate ice and form in-situ cirrus clouds. (b) Experimental setup mimicking the aviation soot atmospheric processes shown in (a). Arrows show the direction of the aerosol flow. See text for details. CS = catalytic stripper; SMPS = scanning mobility particle sizer; DMA = differential mobility analyser; CPC = condensation particle counter; HINC = horizontal ice nucleation chamber; TEM = transmission electron microscopy; CATZ = cloud aerosol transitioner Zurich; OPC = optical particle counter; CPMA = centrifugal particle mass analyser.

## 3 Results

### 3.1 Ice nucleation measurements

The ice nucleation onsets (defined as AF = $10^{-3}$) of unprocessed and processed aviation soot, are summarized in Fig. 2. Unprocessed aviation soot nucleates ice at or above RH required for the homogeneous freezing of solution droplets ($RH_{hom}$) at 218 and 228 K. CS-soot on the other hand, nucleates ice at a lower RH compared to unprocessed soot and below $RH_{hom}$ for few experiments, giving a larger spread in the onset conditions. Ice nucleation behaviors of unprocessed and CS-soot investigated in Testa et al. (2024) using the same experimental setup show that aviation soot possesses few mesopores required to trigger PCF, however $H_2SO_4$ present in most samples and thought to be condensed in the soot pores, prevents the freezing of the pore water by lowering its homogeneous ice nucleation rate. Upon stripping, $H_2SO_4$ is largely removed and the particles

can nucleate ice via PCF for few experiments, and for others the lack of hydrophilic surface oxygenated functionalities likely prevented pore filling and subsequent ice nucleation. The overall poor ice nucleation abilities of unprocessed and CS-soot was

imputed to their limited mesoporosity due to highly overlapping primary particles. CP-soot investigated in the present study nucleates ice also at $RH_{hom}$ and above at $T = 218$ and 228 K, i.e., its ice nucleation ability is essentially similar to unprocessed soot (Fig. 2). At 218 K only 2 out of 13 engines trigger PCF at 5 % $RH_i$ below $RH_{hom}$ (blue outlier scatter points). $D_m$ for the two engines are 350 and 200 nm, compared to the averaged mode of 250 nm for all experiments. Large soot aggregates have higher chance to include a cavity that can trigger PCF compared to smaller aggregates due to their higher number of primary

particles (Zhang et al., 2020; Nichman et al., 2019; Gao et al., 2022a), which explains the higher ice nucleation activity of the 350 nm particle experiment, but not the 200 nm sample. This implies that aggregate size alone is not a sufficient descriptor to explain the modest ice nucleation ability of the 200 nm diameter soot sample. The CS-CP-soot particles show the lowest median nucleation onset at 7 and 4 % $RH_i$ below $RH_{hom}$ at 218 and 228 K, respectively.

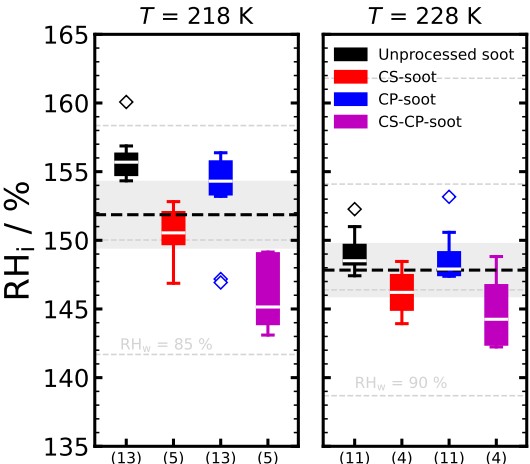

**Figure 2.** Ice nucleation onset $RH_i$ range (AF = $10^{-3}$) of unprocessed and processed soot at the given temperatures. The box extends from 25th to 75th percentiles, the white bar shows the median, the whiskers include the inter-quartile range and the scatter points show the outlier data points. Left and right panels share the same y-axis and legend. The dotted black line indicates $RH_{hom}$ (Koop et al., 2000) and the grey shading its uncertainties in HINC. The $RH_w$ are shown by the dotted grey lines in steps of 5 %. The x-axis shows the number of measurements conducted per sample type (including different engine types).

## 3.2 Soot morphology

### 3.2.1 Mass measurements

The mass of size selected unprocessed and CP-soot in the range 200-400 nm is shown for 4 engines in Fig. 3a and in the range 100-200 nm for the PW4168A engine in Fig. 3b. Fit to mass-mobility relation (Eq. 2) are additionally shown in the figure. The mass-mobility exponent of the unprocessed soot (Fig. 3a) is smaller than measured for engine exit plane aviation soot (about 2.20-2.70 for soot emitted at low to high thrust; Abegglen et al., 2015; Durdina et al., 2014). This is because the particles sampled in our study undergo coagulation in an aerosol reservoir, as a result, the particles become larger and more lacy, which translate to smaller mass-mobility exponent. This is also visible when comparing images of small and compact uncoagulated with unprocessed coagulated soot particles on Figs. A1 and A2, respectively. As expected, the mass-mobility exponents for the smaller PW4168A soot is higher ($D_{fm} = 2.24$, Fig. 3b) and approach values measured for engine exit plane aviation soot. For the engines shown in Fig. 3a, the soot aggregate mass increases for given sizes, which we explain by particle densification through its compaction, resulting in a higher mass-mobility exponent. For the smaller PW4168A engine soot particles ($D_m = 150$ nm; Fig. 3b), we also note a mass change upon contrail processing although within the measurement uncertainties. Despite particles passing through the drier before entering the CPMA, part of the increase in aggregate mass upon contrail processing might be due to water retained in soot aggregate, that do not have the time to desorb during the short residence time in the drier (Gao and Kanji, 2022b). From our data-set, we cannot differentiate between mass increase from particle restructuring and water taken up and thus mass-mobility exponents for the CP-soot samples represent higher estimates.

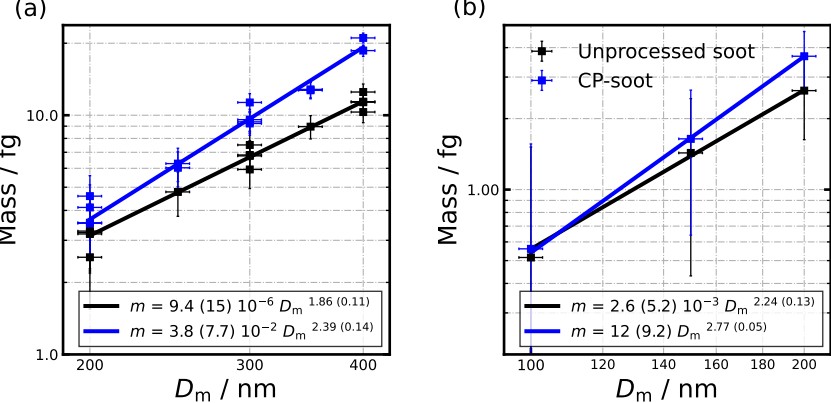

**Figure 3.** Unprocessed and contrail processed size-selected aggregate mass mode ($m$) as function of their electric mobility diameter ($D_m$) for (a) the CFM56-7B26/3, CFM56-5B4/P, CFM56-7B27/3 and CFM56-7B24/3 engines and (b) a PW4168A engine. Lines show fits to Eq. 2 with retrieved parameters shown in the boxes. Uncertainties on the parameters are shown in parenthesis. (b) shares its legend with (a)

### 3.2.2 Size measurements

Differences in $D_{\mathrm{m}}$ between unprocessed and processed soot ($\Delta D_{\mathrm{m}}$) is shown in Fig. 4 for the PW4168A, CFM56-7B26 and
145 PW4062A-3 engines. $D_{\mathrm{m}}$ are derived from log-normal fit to size distribution measured by a first SMPS (SMPS1 in Fig. 1b) for unprocessed soot, the latter monitoring soot aggregate size change due to coagulation in the aerosol reservoir and from a second SMPS (SMPS2 in Fig. 1b) measuring unprocessed and processed soot sizes in parallel with the ice nucleation measurements. Both SMPS have been calibrated against a reference SMPS.

CS-soot size is essentially similar to the unprocessed soot ($\Delta D_{\mathrm{m}}$ within measurement uncertainties) for all engines. For CP-
150 and CS-CP-soot, $D_{\mathrm{m}}$ is decreased for all engines but within measurement uncertainties for the PW4168A engine. We note that although the difference remains within measurement uncertainties, $\Delta D_{\mathrm{m}}$ is consistently larger for CP-soot compared to CS-CP-soot (as confirmed in Sect. 3.2.3). This would be expected as soot particles become more hydrophobic upon catalytic stripping, hence less sensitive to compaction upon contrail processing (discussed further in Sect. 4).

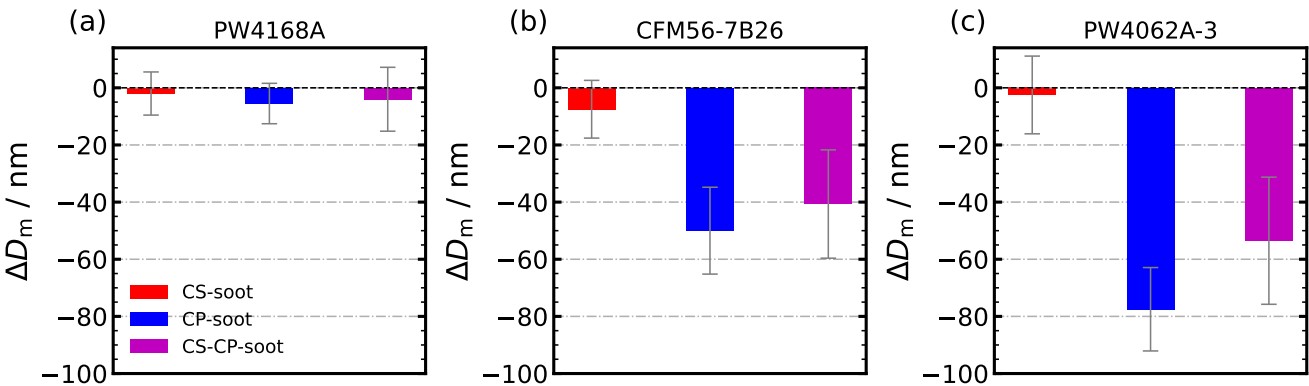

**Figure 4.** Mode diameter change $\Delta D_{\mathrm{m}}$ upon processing of the particles as measured by SMPS1 and SMPS2 (Fig. 1b) for the given engines.

### 155 3.2.3 Shape analysis from TEM

The shape analysis from the TEM images is shown for 3 engines in Fig. 5. As expected, for both unprocessed and contrail processed particles, the convexity increases for decreasing $L$, meaning that smaller soot aggregates are more compact than larger ones. Overall, contrail processing induces a strong increase in convexity and reduction in $L$ of the soot particles for all investigated engine types, which is indicative of aggregate compaction (China et al., 2015) (exemplary TEM images are shown
on Fig. A2). The convexity of the CP-soot does not seem to be correlated with the unprocessed soot size, i.e., median $L$ values for the unprocessed samples of the investigated engines are 190, 515 and 689 nm, resulting in particle convexity after contrail processing of 0.81, 0.81 and 0.78 (boxplots in Fig. 5a-c, respectively). This means that on average, the soot particles become similarly compacted regardless of their initial sizes. This also indicates that the relative increase in compaction is stronger for initially larger aggregates which are more lacy. For instance, the convexity increased by 60 % for 689 nm (median $L$) sized

PW4062A-3 soot and by 10 % for 190 nm sized PW4168A soot. We note that the size reductions measured by the SMPS is smaller than measured from the TEM images. A strict comparison with the TEM measurements is not possible nor expected because of the different assumptions in electrical mobility sizing and the 2D images used for TEM sizing. Nonetheless, the TEM analysis overall corroborates the size and mass measurements (Sect. 3.2.1 and 3.2.2). Upon contrail processing, aggregates mass at given sizes increase. This together with a decrease in particle size as revealed by SMPS and TEM images indicate an increase in particle density caused by the compaction of the aggregates, which become less hollow.

CS-soot convexity and $L$ (boxplots in Fig. 5a-c) are similar to unprocessed soot, i.e., its morphology does not change upon stripping. For the 3 engines investigated, CS-CP-soot show morphology changes similar to CP-soot, i.e., decrease in $L$ and increase in convexity in comparison to the unprocessed soot. However the CS-CP-soot undergo a smaller size reduction than the CP-soot (see also SMPS measurements in Fig. 4) and are slightly less convex (most pronounced for the PW4062A-3 engine). Soot aggregates activated in the contrail chamber undergo similar compaction. Yet, aggregates that did not activated as ice crystals in the chamber (about 80 % at 105 % $RH_w$ for both PW4062A-3 CP- and CS-CP-soot; Fig. 6c) might still undergo hygroscopic growth, depending on their hydrophilicity, leading to their partial collapse (Pagels et al., 2009; Mahrt et al., 2020b). Ice crystal soot residuals and interstitial soot were both imaged with TEM. The lower convexity and size reduction for CS-CP-soot compared to CP-soot indicate that interstitial CS-CP-soot are less compacted than interstitial CP-soot and that CS-CP-soot are on average more hydrophobic. This is due to the removal of $H_2SO_4$ upon catalytic stripping of the particles (Testa et al., 2024). The weaker morphological change for the PW4062A-3 CS-CP-soot could be imputed to a lower content of surface polar groups or higher surface graphitization (Persiantseva et al., 2004; Popovicheva et al., 2011) compared to the other engines investigated, which resulted in its lower convexity.

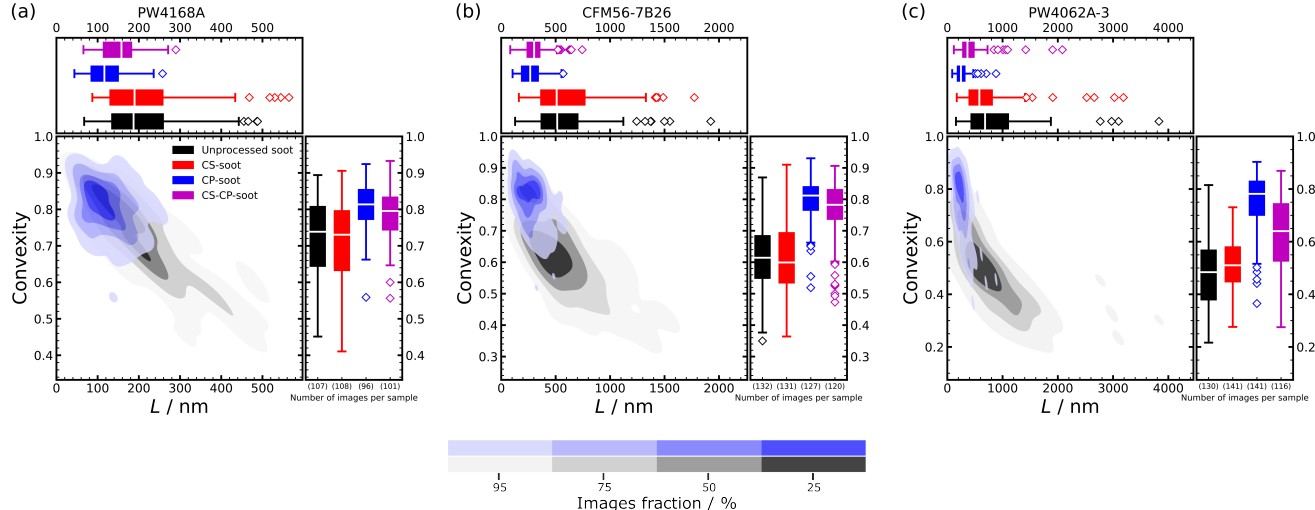

**Figure 5.** Maximum 2D projected length (*L*) and convexity of unprocessed and processed aviation soot aggregates analyzed from TEM images for 3 engines. Convexity and *L* distributions are shown for all soot populations in the boxplots (box statistic as in Fig. 2) and the density contours in the central figures only for unprocessed and CP-soot for clarity. The colored areas enclose -from dark to light- about 25 %, 50 %, 75 % and 95 % of the images (one aggregate per image). The number of images analyzed are shown on the x-axis below the boxplots. (a) shares its legend with (b) and (c).

## 4 Discussion

The inability of CP-soot to promote ice nucleation at RH < $RH_{hom}$ (Fig. 2) indicates that contrail processing does not generate pores relevant for PCF despite compaction of the soot particles as revealed with imaging, size-distribution and size-resolved particle mass measurements. Gao and Kanji (2022a) observed moderate ice nucleation enhancement for contrail processed 200 nm propane soot coated with $H_2SO_4$ compared to unprocessed coated soot. The authors explain it by the presence of $H_2SO_4$ condensed in pores and redistributing over the newly formed pore network upon compaction, hence limiting the ice nucleation of the pore water. The same argument might explain the inability of contrail processed aviation soot to promote ice nucleation via PCF. CP-soot ice nucleation AF for 3 engines are shown in Fig. 6 (the associated soot size distributions are shown in Fig. B1) and have all onset at $RH_{hom}$. We note a small ice nucleation enhancement for the CFM56-7B26 engine after contrail processing that we attribute to the low soot sulfur content for this engine (0.02 atomic % measured with TEM X-ray spectroscopy; Testa et al., 2024).

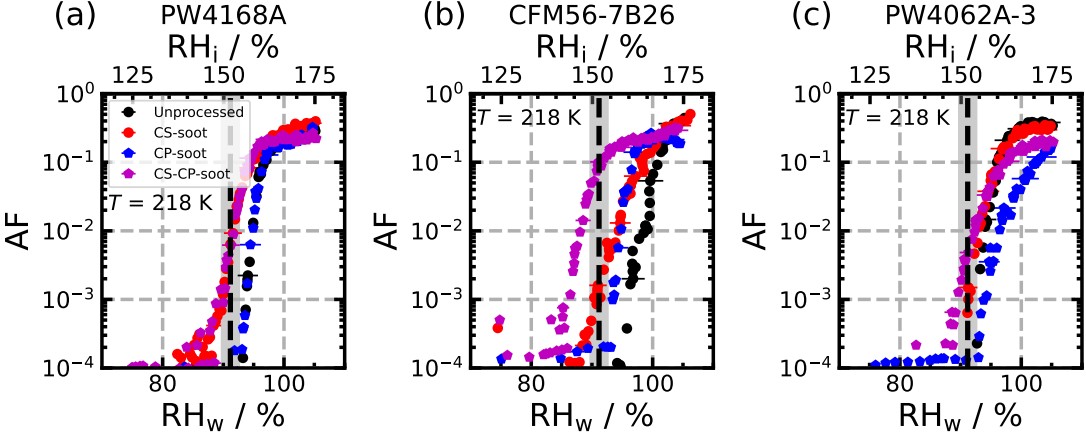

**Figure 6.** AF ice nucleation curves as a function of RH at 218 K for the given engines. The dotted black line and grey shading indicate $RH_{hom}$ and associated uncertainties. (a) shares its legend with (b) and (c).

As mentioned above, the ice nucleation enhancement upon catalytic stripping (Fig. 2) is attributed to the removal of $H_2SO_4$ (and organics) condensed in pores while stripping, emptying existing mesopores that become available for water condensation and ice nucleation (Testa et al., 2024). Yet, the ice nucleation response to stripping varies between the 3 engines investigated in Fig. 6. For the PW4062A-3 soot sample, stripping does not change the ice nucleation AF, presumably due to its hydrophobic surface indicated by the weak morphological change of the corresponding CS-CP-soot sample (convexity = 0.64; Fig. 5c). CS-soot from the PW4168A and CFM56-7B26 engines trigger ice nucleation at a lower RH compared to unprocessed soot, and clearly below $RH_{hom}$ only for PW4168A CS-soot. Better soot water uptake capacity for those engines would corroborate with their ice nucleation abilities. Differences in primary particle arrangement and overlap could also contribute to the small difference in ice nucleation between CS-soot from the PW4168A and CFM56-7B26 engines (exemplary images shown in Fig. A2). A more detailed analysis of primary particle sizes and overlap would be needed to be quantitative on the effect of morphology.

For the CS-CP-soot (Fig. 2), the aggregate compaction upon contrail processing would have increased the number of empty and therefore PCF-relevant mesopores due to formation of new cavities between the soot primary particles (Mahrt et al., 2020b; Gao et al., 2022a). Nonetheless, the ice nucleation onset of the CS-CP-soot is close to $RH_{hom}$ and they remain much weaker INPs than soot used as aviation surrogates (Mahrt et al., 2020b, a; Gao et al., 2022a; Gao and Kanji, 2022a), despite similar or stronger compaction of aviation soot (median circularity = 0.25 in Mahrt et al., 2020b, and 0.39 in this study; Table A1). For PCF to be triggered by soot, the particles need to possess cavities with the right size and shape. Such cavities should be found for soot primary particles diameter ($D_{pp}$) of 10 to 30 nm with moderate overlaps (Marcolli et al., 2021). Smaller primary particles need to be close to point contact to give rise to cavities that can accommodate the ice germ, and larger primary

particles need to strongly overlap to give rise to small enough cavities than can be filled below water saturation. However, at any diameter, the cavities close if primary particles are too fused, inhibiting PCF. Images of the soot particles sampled in this study (Supplementary Figs. A1 and A2) reveal that the primary particles are highly fused (strong overlap) and that $D_{\mathrm{pp}}$ ranges from 10 nm to 90 nm in a single aggregate. This explains why the CS-CP-soot has limited ice nucleation ability upon compaction. Small primary particles might fill the cavities formed between larger ones, reducing the potential for pore volume generation. A combination of the highly fused nature of primary particles and the large $D_{\mathrm{pp}}$ range together with poor water uptake capacity at low RH (Testa et al., 2024) contribute to the overall weak ice nucleation abilities of the CS-CP-soot. Nonetheless, contrail processing of the CS-soot leads to different response as shown by the large spread of ice nucleation onset (Fig. 2) and by the AF curves shown in Fig. 6. CS-CP-soot particles from the PW4062A-3 engine are unable to promote PCF despite their large sizes (Figs. 5c and B1c). We attribute this to their poor water affinity and moderate compaction (convexity = 0.64). In contrast, both PW4168A and CFM56-7B26 CS-CP-soot are similarly compacted (convexity about 0.79). However, only soot from the latter promotes clear PCF. For the PW4168A engine, CS-CP-soot essentially nucleates ice similar to its CS-soot sample. The small number of primary particles per aggregate due to small aggregate sizes (median $L$ = 190 nm) may limit the number of mesopores generated upon contrail processing. CFM56-7B26 soot aggregates are twice the size on average, giving rise to higher probability of generating cavities and inter-aggregate voids relevant for PCF upon compaction.

## 5    Atmospheric implications and limitations

Recent measurements (Testa et al., 2024) showed that aviation soot is unlikely to promote ice nucleation below conditions required for homogeneous freezing of solution droplets ($RH_{\mathrm{hom}}$), particularly because the sizes of emitted soot particles are below 100 nm, which have been shown to be poor INPs for all conditions (variety of mixing states and particle morphologies). Here, we show that contrail processed aircraft turbine engine soot particles (not proxies) remain poor INPs at 218 and 228 K despite a strong compaction of the particles upon contrail processing, thought at first to be the reason for ice nucleation enhancement. This supports the results from Kärcher et al. (2021, 2023) who estimated that only a small fraction of aviation soot nucleates ice at cirrus atmospheric conditions, leaving cirrus cloud properties essentially unperturbed. Due to constraints of the measurement facility in this work, the representation of aircraft exhaust processes in the ground set up and hence of the aviation soot properties might differ from those at flight altitude. For instance, differences in equilibrium temperatures and dilution would impact the partitioning of volatile unburned hydrocarbons and sulfur compounds and hence the soot ice nucleation ability. The higher temperatures in our setup (433 K, then room temperature, Sect. 2.1) compared to upper tropospheric temperature (< 228 K; Krämer et al., 2020), would not promote the condensation of volatiles onto soot as much as would occur at flight altitude temperatures. The interaction of nucleation mode particles with aviation soot in the young aircraft plume downstream of the engine is thought to increase the soot coating (Kärcher et al., 1998; Yu et al., 1999; Kärcher et al., 2007), but does not occur in our ground setup due to the absence of nucleation mode particles. Thus, soot particles in this study are expected to have a lower amount of coating from this effect. On the other hand, total particle surface area was less in our ground setup

due to the absence of the nucleation mode particles, and the exhaust gases experience reduced dilution in the aerosol reservoir with synthetic air, compared to the strong dilution that would occur within the first seconds at flight altitudes (Kärcher et al., 2007). This effect would enhance the condensation of volatiles onto the soot particles in our ground setup. However, even if lower amounts of organics and sulfate condense onto the soot particles in the atmosphere, these would first condense into the pores of the soot, due to the capillary effect, and thus inhibit ice nucleation of the soot particles. Thus, our conclusions of

the poor ice nucleating ability of contrail processed soot would remain the same. Downstream of HINC1, the reemitted soot aggregates get compacted due to contrail processing and their sizes decrease. We note that, the convexity/size change of the contrail processed aviation soot sample from this study is very similar to what measured for other cloud/contrail processed soot types (Ma et al., 2013; China et al., 2015; Mahrt et al., 2020b). This is explained by the harsh activation process occurring in HINC1 and CATZ (activation in cloud droplets and ice crystals followed by sublimation; Corbin et al., 2023). Although large

compaction was observed in our study (convexity approaching 1), we do not rule out that smaller condensation/sublimation rate in the atmosphere could lead to larger compaction of the particles. The formation of large soot aggregates (Petzold et al., 1998, 1999) due to the coagulation between ice crystals and scavenging of interstitial soot aggregates was not possible due to the low concentration of ice crystals and soot and non-turbulent flow in HINC1. Due to the absence of nucleation mode particles, coagulation of these with ice crystals, and coagulation with interstitial soot aggregates was not possible in HINC1.

However, the absence of these processes does not change our conclusions as adding more organics onto the soot particles would only further result in poor ice nucleation activity (Testa et al., 2024; Gao and Kanji, 2022a). Gas phase chemistry and particle oxidation are thought to considerably slow down while exiting the combustor chamber due to much lower temperatures in the exhaust nozzle and downstream of the engine (Wong et al., 2008; Dakhel et al., 2007). Such a drop in temperature was also present in our sampling system (thousand degrees to 433 K and to room temperature), thus the primary particle overlap, size,

crystallinity, and oxidation should be unaffected and comparable to in situ emitted aviation soot particles.

Summarizing, soot particles in our ground setup were larger and less dense than in situ soot due to coagulation in our aerosol reservoir. We believe the aggregate compaction in this work is atmospherically relevant as the parameters driving the soot compaction, i.e., $RH_i$ experienced by the particles and bulk water condensation were represented in our ground setup. The primary particle properties and oxidation are fixed in the combustor and hence should be representative of their in-situ

counterpart. Finally, we expect in situ particles to be coated with $H_2SO_4$ and organics but to which extent the coating of the aviation soot sample in our study is different from in situ aviation soot cannot be established from our study. For this reason, we quantified the ice nucleation ability of coating free (CS-CP-soot) and coated (unprocessed and CP-soot) soot in our study to constrain the possible effect of different soot mixing states on aviation soot ice nucleation.

Aviation soot samples that were catalytically stripped and contrail processed were able to nucleate ice around 145 % $RH_i$

at 218 K ($\sim$7 % lower than $RH_{hom}$). The modest ice nucleation ability for the CS-CP-soot likely arises from increased cavity number and sizes within the soot aggregates, which would be absent in the unprocessed soot samples coated with organics and sulfate. We note that, if glassy organic coatings formed on the contrail processed (and unprocessed) soot particles, these particles may form ice by deposition nucleation (Knopf et al., 2018), which was not observed due to the high ice nucleation onset above $RH_{hom}$. Thicker coatings for flight altitude aviation soot compared to our CP-soot would favor homogeneous

nucleation and thinner coatings would be bound by the ice nucleation ability of our CS-CP-soot sample. Nevertheless, as long as aviation soot is co-emitted with $H_2SO_4$, it is likely to acquire a coating upon emission and further in the exhaust plume (Kärcher et al., 2007), thus we expect our unprocessed and CP-soot samples to be of higher atmospheric relevance for engines and fuel currently in use.

Additionally, we show that aggregate size does not predominantly regulate the ice nucleation of contrail processed aviation
soot but rather the differences in $H_2SO_4$ coating and primary particles properties. However, although large aggregates do not necessarily promote PCF upon contrail processing, we stress that very small aggregates can nonetheless inhibit PCF. The PW4168A experiment presented soot with the smallest sizes among our experiments ($D_m < 150$ nm) shows that the effect of contrail processing on the ice nucleation ability remains limited. PW4168A soot gets compacted upon contrail processing (convexity = 0.8) but the aggregates possess too few primary particles due to their small sizes to generate PCF-relevant pores
upon compaction. Furthermore, we emphasize that relevant sizes for aviation soot are a lot smaller (only 0.2-2 % in number are larger than 150 nm, Fig. B1), we thus expect contrail processing to be further limited for in situ emitted aviation soot as for our PW4168A sample.

The number and size of cavities within the soot aggregates are the primary controlling factors of soot ice nucleation via PCF (Sect. 1). The cavity formation is controlled by the primary particle morphology (being determined in the engine combustor and
therefore well simulated in our ground setup) and the aggregates size. High overlap of the primary particles has been observed on soot samples for all tested engines. Smaller aggregate size for aviation soot is expected for turbofan engines, hence we believe that the results from our study, that is, aviation soot particles are poor ice nucleating particle for cirrus formation can be generalized to soot emitted with the current aircraft fleet and fuel (Jet A/A-1; > 90 % of global usage; Jing et al., 2022; Pires et al., 2018).

In addition to contrail processing, several soot aging processes can occur in the atmosphere, such as interaction with background aerosols and volatile compounds, or oxidation of aerosols by $O_3$ and OH radicals (Bond et al., 2013). Interception of soluble aerosol onto the soot surface would increase the amount of soot coating preventing PCF. Exposure of aviation soot to $O_3$ or OH at flight altitude can cause condensed organics to desorb due to the breaking of covalent bonds with the elemental soot fraction. The oxidized organics could recondense onto the soot due to their lowered volatility, e.g., short alkanes and alde-
hyde that are soluble in acidic solution (e.g., $H_22SO_4$ solution; Yu et al., 1999) and could lead to a freezing point depression if condensed in pores or prevent water uptake by blocking the pores if there are hydrophobic, inhibiting PCF (Gao and Kanji, 2024). Nevertheless, we do not rule out that oxidation of soot surface organics can also turn into glassy coatings and promote deposition ice nucleation of the particles (Tian et al., 2022).

Additionally, the presence of other potent atmospheric INPs would further limit the effect of aviation soot on cirrus cloud
microphysical properties. For instance, we note that the CS-CP-soot $RH_i$ onset remains substantially above that for mineral desert dust, e.g., about 120 % (Ullrich et al., 2017), which all outcompete soot ice nucleation.

An unknown for the near future is alternative aviation fuel (Kärcher, 2018), mainly synthetic fuel and biofuel, used as pure or blended with jet fossil fuel (Burkhardt et al., 2018; Voigt et al., 2021). Pure synthetic and biofuel are free of sulfur and aromatics (Braun-Unkhoff and Riedel, 2015), hence soot emitted from pure alternative fuel or a blending with jet fossil fuel will

have different properties than jet fossil fuel soot, therefore affecting their ice nucleation abilities. Changes in soot morphology and nanostructure have been suggested for different types of alternative fuel (Lobo et al., 2016; Liati et al., 2019; Huang and Vander Wal, 2013; Vander Wal et al., 2022). Liati et al. (2019) found alternative fuel soot to be generally more graphitized (= more hydrophobic; Haul, 1982; Popovicheva et al., 2008) than jet A-1 fuel soot. The same authors also observed blended fuel soot associated with an amorphous organic outer-shell, potentially increasing or decreasing the soot water uptake capacity,

depending on the nature of the organics. On the contrary, Trueblood et al. (2018) found no change in particle hydrophilicity at cruise thrust between conventional and alternative fuel. Although more such studies would be needed to be quantify the properties of alternative fuel soot, the ice nucleation abilities of sulfur free aviation soot can be hypothesized from our study. Those might correspond to our CS-CP-soot sample, free of sulfur and organics as discussed above and which exhibits moderate ice nucleation ability, particularly in comparison to mineral dust. We note that aviation soot particles emitted from alternative

fuel are thought to be on average smaller and emitted at lower concentrations at all sizes (Durdina et al., 2021; Moore et al., 2015, 2017; Liati et al., 2019; Lobo et al., 2016). We expect both properties to further limit their effect on cirrus clouds.

## 6   Conclusions

This study presents ice nucleation measurements and morphology analysis of polydisperse aviation soot particles sampled from in-use commercial aircraft engines, with a focus on the effect of contrail processing on aviation soot ice nucleation

ability. The effect of coating on the ice nucleation ability and morphology of contrail processed soot was investigated by catalytically stripping the particles at 623 K, removing volatile organics and sulfur. The ice nucleation ability of unprocessed and processed particles was assessed with a continuous flow diffusion chamber operated at temperatures relevant for cirrus cloud formation ($T \leq 228$ K) and varying $RH_i$ (110-170 %). Particle morphology was investigated with electron microscopy, aerosol sizing and mass measurements. The measurements reveal that contrail processed aviation soot particles are poor INPs

forming ice at RH required for homogeneous freezing of solution droplets ($RH_{hom}$) despite contrail processing inducing strong compaction of the soot aggregate (e.g., aggregate convexity increased up to 65 % and maximal aggregate length decreased similarly). Results for the catalytically stripped and catalytically stripped plus contrail processed soot suggest that the presence of $H_2SO_4$/organics condensed in the soot aggregate cavities prevent the particles from promoting ice nucleation via PCF. After catalytic stripping, large aviation soot aggregates are able to promote PCF likely due to formation of new aggregate

cavities and void upon compaction but still required RH close to $RH_{hom}$. Limited ice nucleation enhancement for catalytically stripped contrail processed aggregate is likely due to the soot primary particles being highly fused and their large size range over single aggregate hence limiting the generation of large cavities upon compaction. Analysis from microscopy images shows that aggregates as small as 150 nm mode diameter (corresponding to 0.2-2 % of atmospheric relevant aviation soot size distribution) get compacted upon contrail processing but that contrail processing remain inefficient in promoting PCF for the

catalytically stripped particles due to their small sizes. Overall, our results suggest that aviation soot particles would likely not serve as INP for cirrus formation and that current radiative forcing estimates (Lee et al., 2021; Righi et al., 2021) should be updated. Extrapolation of our results to soot emitted from alternative jet fuel also suggests that their ice nucleation activity

will likely remain negligible. Other effects from alternative jet fuel or new engine design such as lower soot emission index on contrail formation remain however unquantified.

*Data availability.* The data for the figures in this work are available at DOI: https://doi.org/10.3929/ethz-b-000649178

## Appendix A: TEM images of unprocessed and contrail processed aviation soot particles

Aviation soot samples were collected for different engine types and imaged with TEM. In Fig. A1, exemplary images are shown for particles collected downstream of the test cell similar as Testa et al. (2024). Those aggregates undergo no coagulation and presumably no morphological change before sampling. The length of primary particles that are unambiguously identifiable are indicated for two images on the Figure. The primary particle diameters range from 17 to 64 nm for those images with features close to 100 nm that could correspond to several fused primary particles. Exemplary images of unprocessed and contrail processed soot aggregates for 3 engine types are shown in Fig. A2. Median shape parameters retrieved from the TEM images of unprocessed and processed aggregates are shown in Table A1.

## Non coagulated soot aggregates

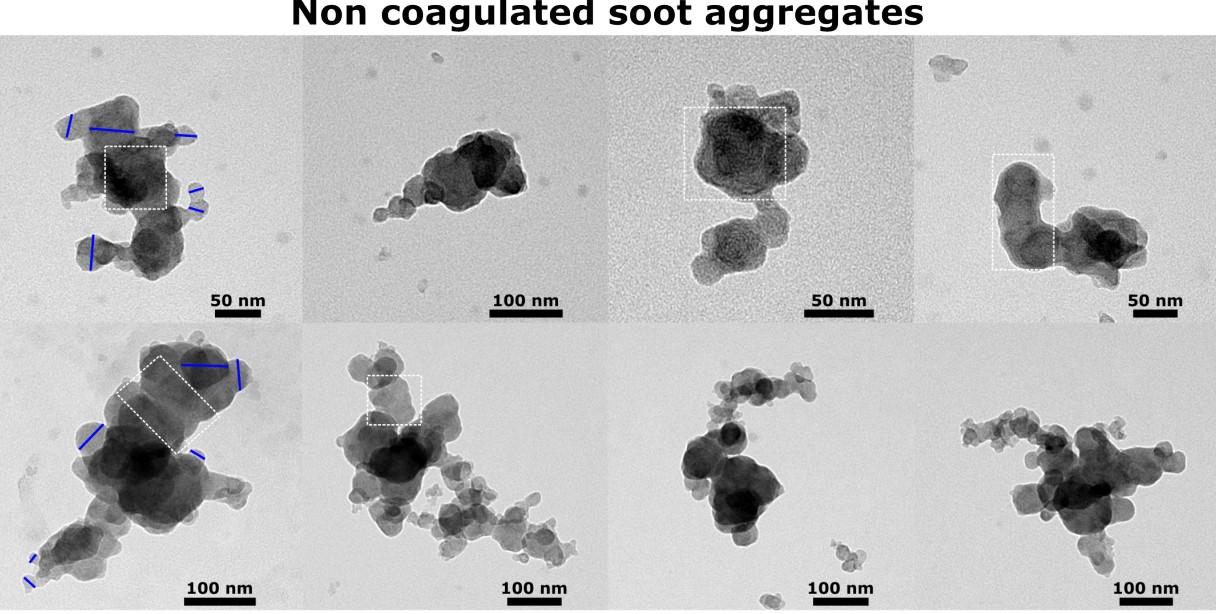

**Figure A1.** TEM images of soot aggregates collected downstream of the test cell and undergoing no coagulation. The blue lines on two of the images highlight the length of identifiable primary particles. Regions highlighted with dotted white rectangles show fused primary particles. Note the different scale bars in the images.

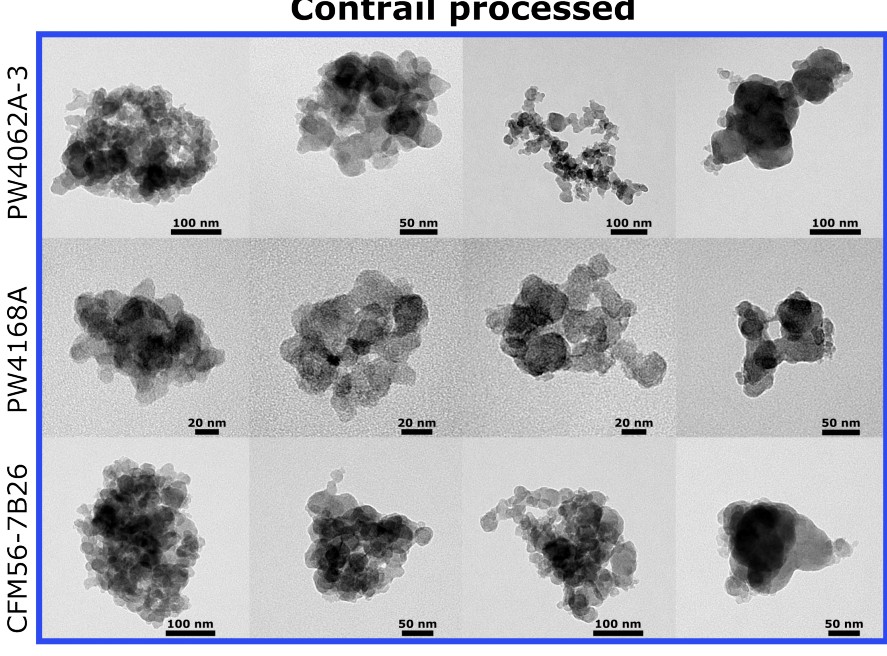

**Figure A2.** TEM images of unprocessed and contrail processed soot aggregates for the given engines. Note the different scale bars in the images.

**Table A1.** Aviation soot shape parameters derived from TEM images for unprocessed and processed particles. Numbers represent median values for 4 engine types (CFM56-7B26/3, CFM56-7B26, PW4168A and PW4062A-3). One standard deviations are shown in parenthesis. $L$ = 2D-projected maximal aggregate length; $W$ = 2D-projected aggregate width; $D_{\mathrm{2D,eq.}}$ = 2D-projected equivalent spherical diameter.

| Soot sample | Convexity | Circularity | Aspect ratio | $L$ / nm | $W$ / nm | $D_{\mathrm{2D,eq.}}$ / nm |
|---|---|---|---|---|---|---|
| Unprocessed | 0.61 (0.09) | 0.22 (0.04) | 1.51 (0.05) | 490 (179) | 330 (119) | 283 (81) |
| CS-soot | 0.6 (0.09) | 0.22 (0.04) | 1.45 (0.05) | 509 (170) | 365 (113) | 296 (84) |
| CP-soot | 0.8 (0.04) | 0.39 (0.03) | 1.34 (0.04) | 257 (100) | 194 (77) | 183 (67) |
| CS-CP-soot | 0.78 (0.07) | 0.37 (0.06) | 1.35 (0.03) | 302 (86) | 229 (70) | 209 (55) |

# Appendix B: Aviation soot size distributions

Fig. B1 shows soot size distributions from the PW4168A, CFM56-7B26 and PW4062A-3 engines (measured with the SMPS2 as shown in Fig. 1b). Aviation soot particles measured in situ (CFM56-2-C1 engine fueled with Jet A-1 fuel at medium thrust; Moore et al., 2017) and at ground level (CFM56-7B26 engine fueled with Jet A-1 fuel at 50-65 % see level thrust; Durdina et al., 2021) are shown for comparison. Coagulated particles sampled in this study present narrowed size distributions and shifted to larger sizes compared to uncoagulated particles. Considering that the larger particles nucleate ice at the lowest RH, the largest 0.1 % soot particles likely contributes to the ice nucleation onsets discussed in the main text (Figs. 2 and 6). The cumulative fraction distributions reveal that the 0.1 % of the larger aerosol in our experiments are considerably larger than majority of aviation soot sizes measured by Moore et al. (2017) and Durdina et al. (2021). The largest 0.1 % for both CFM56-7B26 and PW4062A-3 engines are >900 nm, and >450 nm for the PW4168A engine. As a comparison, the largest 0.1 % of in situ aviation soot are >200-300 nm.

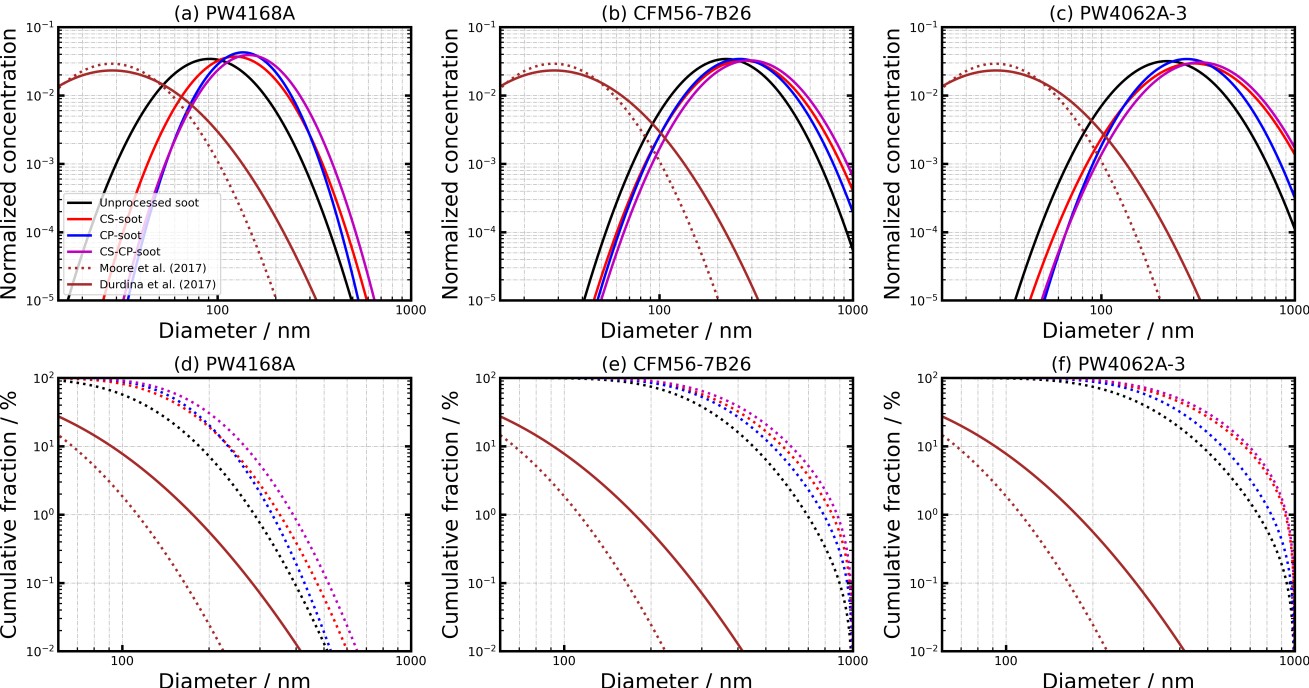

**Figure B1.** (a-c) Normalized aviation soot size distributions of unprocessed and processed particles for 3 engine types. Aviation soot size distributions measured in situ (Moore et al., 2017) and at at ground level (Durdina et al., 2021) are shown for comparison. The same size distributions are shown as cumulative fraction in (d-f).

*Author contributions.* The experiments were designed by BT and ZAK with help from LD, JE and CS. BT conducted the experiments and performed the data analysis. BT wrote the first draft and all authors contributed to data interpretation and writing of the manuscript. ZAK supervised the study, conceived the idea and obtained funding.

*Competing interests.* The authors declare that they have no conflict of interest.

*Acknowledgements.* The work has been supported by the European Commission via their Horizon 2020 Research and Innovation Program under Grant Number 875036 (ACACIA project) and the Swiss Federal Office of Civil Aviation, SFLV-2020-080. The authors declare no conflicts of interest relevant to this study. The authors would like to thank the engine operators of the SRT facility for their help and support. We are also grateful to Fabian Mahrt from Laboratory of Atmospheric Chemistry (PSI) for helpful discussion of the data. We thank Prof.

Pratsinis from the Departement of Mechanical and Process Engineering (ETH) for lending a CPC during our measurement campaign and Prof. Markus Ammann from Laboratory of Atmospheric Chemistry (PSI) for lending the Partector TEM sampler.

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
