# Peer review of "Simulated contrail processed aviation soot aerosol are poor ice nucleating particles at cirrus temperatures"

_EGUsphere, 2024_

## Author Comment (AC1)

**Referee comments are marked in black bold and are numbered as R1Cx with x the comment number.** Author (AR) responses are marked in black directly below the comments. The original text from the manuscript is repeated in blue and corrected text in revised manuscript is typed in red. The former line numbers are given in "()" followed by the new line numbers.

**This reviewed paper is an accompanying paper of "Soot aerosol from commercial aviation engines are poor ice nucleating particles at cirrus cloud temperatures" (doi: 10.5194/egusphere-2023-2441), which is still under review at the time of submission of these comments. Testa et al., in this follow-up study of aircraft-engine emitted soot aerosol, describe the reactivation ability of contrail-processed soot. The experiments were conducted on the ground with 2 HINC ice nucleation chambers, which simulated two consecutive steps of cloud formation with a sublimation step between them, at a range of cirrus-relevant temperatures and RHi. This study claims to advance the current knowledge in the indirect climate effects of soot emitted from jet engines, which are responsible for anthropogenic direct injection of fresh soot into higher altitude. Using real engines, the authors have demonstrated an improvement of the jet combustion soot simulation, in comparison to previous studies conducted with soot surrogates. The authors conclude that jet-engine cloud-processed combustion-soot in its compact form is poor INP, similar to activation of the unprocessed soot, despite the expected increase in activation via the PCF mechanism, so heavily reported in previous studies. The authors underline the importance of sulfuric acid and organic volatiles presence in the activation process and their inhibiting impact. The paper is well written and structured in a logical way.**

**I don't have much comments on the writing or presentation quality but I do have major concerns about the scientific significance and the implied generalized representation of real contrail processes in this and the accompanying paper, both fail to recognize the limitations of such ground experiments even though some of the major uncertainties are mentioned. While such steady flow studies might be relevant for natural cirrus formation, I find there are still major gaps for proper simulation of contrail formation process on the ground.**

**This accompanying paper focuses on the reactivation however, the evaluation of this sequential study can't be done in isolation ?and it is impossible to avoid the review of the initial soot generation and sampling methodology. It must be done in order to properly evaluate the conclusions and the usefulness of the presented results for future contrail research.**

**Essentially this study dismisses the relevance of PCF ice to contrails (many of the earlier studies cited in the manuscript), implying that contrail formation processes shouldn't be discussed anymore in the context of PCF. This is a strong claim, which requires a deeper substantiation and discussion of the limitations of this study and further clarification on the applicability of this study to actual contrails. To demonstrate the relevance of this study to high altitude contrails, the authors should have a more detailed description of timescales and rates of processes in their experiment with a comparison to rates known from real contrails.**

We thank the reviewer for their comments and address each comment individually below. Here we would like to emphasize that indeed PCF is not relevant for contrail formation. The reason being that PCF is relevant for conditions below water saturation in the atmosphere. Contrails form in the wake of aircraft because of an abundant supply of water vapour that results in condensation and droplet formation

followed by homogeneous freezing. PCF has no role to play in this process, it is entirely driven by the availability of excessive amounts of water vapour that readily condense as bulk droplets onto the available soot particles (Kärcher, 1998) and the low temperatures that promote freezing of these droplets. In this regard, we do not make any changes to the manuscript.

**Below I listed some major comments, unanswered questions, and some minor comments. Thus, after a major revision is complete, I'd recommend this manuscript for publication. Please see my comments below:**

**Major comments:**

**R1C1.1: The current design involves a stirred tank for offline characterization with isokinetic flow instruments and filter sample collection. I think the experiments could have been designed better to have a real-time sampling, similar to what was done by Korhonen et al. 2022 however, this would not solve all the problems either.**

AR1.1: Real-time sampling was conducted in our study for aerosol size distribution (with SMPS) and aggregate mass (with CPMA) analysis. Offline sampling in this study and in Testa et al. (2024) was conducted for STXM/NEXAFS and electron microscopy analysis. The former gives information on the fine chemical structure of the soot aggregates, which cannot be easily determined with mass spectrometer (real-time sampling method used in Korhonen et al., 2022). Electron microscopy analysis of the soot aggregates gives shape information that cannot be obtained other than with microscopy techniques that required sampling and offline analysis of the samples.

**R1C1.2: Additionally, I think expansion chambers (e.g. Crawford et al. 2011). might be more representative of the actual dynamic process in contrails. For the current ground setup, there should at least pictures, simulation of the exhaust plume, description of the ambient conditions- was the experiment conducted outdoors, wind/temperature/humidity information, was the heat-resistant alloy probe, located ~1 m downstream of the engine exhaust nozzle, sampling from the middle of the plume, at an angle, etc.**

AR1.2: We agree that our setup does not mimic the dynamical processes and rates relevant for contrail formation, e.g., exhaust temperature dropping rapidly (< 1 s) from thousands of degrees to -60 °C and pressure from tens of bars to below 1 bar (Kärcher, 2018). Instead, the time scales are longer in our experiments, the pressure drops from tens of bars in the engine, to ~1 bar (atmospheric pressure in our measurement set up). The temperature also drops in three steps, from the engine temperature to the heated line 160 °C (see below) and then to room temperature followed by a third drop from room temperature to the cloud chamber temperature. Expansion ice nucleation chambers generate supersaturation by decreasing the pressure. While they represent better the dynamics of raising air parcels, they would also fail at mimicking contrail processes (no rapid cooling and dilution of the exhaust or wing vortices formation). Besides, cooling of the particles down to contrail temperature is faster in CFDC than in expansion chambers (Lacher et al., 2017; Möhler et al., 2021) and more representative of in situ rapid aerosol cooling. So, we do have similar temperature drops as would occur in the atmosphere, but they occur over a slightly longer time scale.

In the following, we further describe our ground setup and discuss the limitations of contrail processes in our setup and the impact on soot properties. The ground-based setup used in this study has been described in Testa et al. (2024), as mentioned in the paper (line (41) 45-46). The sampling unit, i.e., the probe sampling the engine exhaust in the test cell (indoor), is described in studies referenced in Testa et al. (2024). We now

also explain it here and add a description to the manuscript (see below). In brief, the air flow in the test cell and hence feeding the engine is at ambient temperature and humidity. Wind speed and sampling unit meet the International Civil Aviation Organization (ICAO) requirements for engine emission measurements. The engine exhaust particles are sampled at the engine exit plane, so not in the exhaust plume (as stated in Durdina et al. (2014) using the same sampling unit: "*Positions of the probes' orifices in the exhaust stream were adjusted such that air-to-fuel (AFR) ratios calculated from the gaseous emissions data agreed within 15% of the AFR from the engine data, and thus they provided a representative exhaust sample.*"). A long trace-heated line (12 m at 160 °C) transports the exhaust to the aerosol reservoir that was in a room next to the engine test cell (Testa et al., 2024). The primary objective of the study is to quantify the ice nucleation ability of contrail processed aviation soot. Thus, the setup was designed to closely mimic the contrail processing of the sampled soot particles (known to mainly affect soot morphology) rather than mimicking contrail formation.

Regarding the setup description, we propose the following edits. Lines (39-44) 40-51 now read: "The experimental setup was designed to simulate targeted atmospheric processes and to mimic the contrail processing of the sampled aviation soot particles (Fig. 1a). Limitations of the ground setup in representing atmospheric processes (e.g., aircraft exhaust evolution, contrail formation) and soot ice nucleation ability are discussed in Sect. 5. Soot particles were sampled from in-use commercial aircraft engines (multiple models from Pratt & Whitney and CFM International) running in an indoor test cell, with air intake at ambient temperature and humidity. The engines were all fueled with Jet A-1 fuel and ran from low to high power (30-100 % sea level thrust). The detailed sampling system is described in Testa et al. (2024) which we briefly describe here. The engine exhaust gas and particles were sampled by a heat-resistant alloy probe ~1 m downstream of the engine exhaust nozzle and directed by a long trace-heated line (12 m at 433 K) to a stirred tank, that was in a room next to the engine test cell and acted as an aerosol reservoir where the soot particles accumulated and coagulated (Fig. 1b). The exhaust temperature drops in three steps, from the engine temperature (thousands of Kelvin) to the heated line temperature (433 K) and then to room temperature followed by a third drop from room temperature to the cloud chamber temperature (< 228 K, see below)".

Line (49-50) 57 now reads: "[…] set to contrail cloud thermodynamic conditions ($T$ = 228 K and $RH_w$ = 105 %, HINC1 in Fig. 1b) allowing […]".

Gas phase chemistry and particle oxidation is thought to considerably slow down while exiting the combustor chamber due to lower temperatures in the exhaust nozzle and downstream of the engine (Dakhel et al., 2007; Wong et al., 2008). Such drop in temperature is also present in our sampling system (thousand degrees to 160°C), thus the primary particle overlap, size, crystallinity and oxidation are thought to be comparable to in situ emitted aviation soot particles. Differences in equilibrium temperatures and dilution would nonetheless impact the partitioning of volatile unburned hydrocarbons and sulfur compounds, i.e., the soot mixing state, and hence the soot ice nucleation abilities. The higher temperatures in our setup (160°C and then room temperatures) compared to the temperature at which the exhaust would be exposed in the atmosphere (- 60°C) would reduce the condensation of volatiles onto the soot due to their higher equilibrium saturation pressures. Higher temperature together with the drying of the exhaust before sampling into the aerosol tank would also prevent the formation of nucleation mode particles. On the other hand, reduced particle surface area (no nucleation mode particles) would favor condensation of volatiles onto the soot particles. Besides, the gases only get slowly diluted by synthetic air in the aerosol reservoir, compared to the strong in situ dilution of the exhaust gases that occur within the first seconds in the atmosphere (Kärcher et al., 2007). Nonetheless, modelling studies (Kärcher, 1998;

Kärcher et al., 2007; Yu et al., 1999) showed that nucleation mode particles and aviation soot are thought to interact in the young aircraft plume downstream of the engine, increasing the soot coating. This process did not take place in our measurement due to the absence of nucleation mode particles. Downstream of HINC1, the reemitted soot aggregates get compacted due to contrail processing and their sizes decrease. However, the formation of large soot aggregates due to the coagulation between ice crystals and the scavenging of interstitial soot aggregates is not possible due to too low concentration of ice crystals and soot and the flow being laminar in the chamber. Next, coagulation of interstitial volatile particles with ice crystals, and coagulation between interstitial soot aggregate with volatile particles is not possible in the chamber due to the absence of nucleation mode particles (although coagulation processes in real contrail remain unquantified today; Moore et al., 2017; Petzold et al., 1998).

Summarizing, we expect in situ particles to be coated with $H_2SO_4$ and organics. To which extent the coating of the aviation soot sample in our study is different from in situ aviation soot cannot be quantified, but any condensation of organics or sulfate in the atmosphere would first condense into soot pores which inhibits ice nucleation (Gao et al., 2022; Gao & Kanji, 2022), and thus our conclusions for the efficiency of aviation soot ice nucleation remain the same. Next, the soot particles are larger and less dense in our setup than in situ soot due to coagulation in our aerosol reservoir, but their compaction should be representative of real contrail compaction (see AR13, AR20). Finally, the primary particle properties and oxidation are fixed in the combustor (see above and AR4) and hence should be representative of their in-situ counterpart. This is described in part in the atmospheric section in Testa et al. (2024), but we now also add it as a discussion in the atmospheric implication section of the current paper.

The title of Sect. 5 has been modified for: "5 Atmospheric implications and limitations"

Lines (238) 242-283 now read: "[…] leaving cirrus cloud properties essentially unperturbed. Due to constraints of the measurement facility in this work, the representation of aircraft exhaust processes in the ground set up and hence of the aviation soot properties might differ from those at flight altitude. For instance, differences in equilibrium temperatures and dilution would impact the partitioning of volatile unburned hydrocarbons and sulfur compounds and hence the soot ice nucleation ability. The higher temperatures in our setup (433 K, then room temperature, Sect. 2.1) compared to upper tropospheric temperature (< 228 K; Krämer et al., 2020), would not promote the condensation of volatiles onto soot as much as would occur at flight altitude temperatures. The interaction of nucleation mode particles with aviation soot in the young aircraft plume downstream of the engine is thought to increase the soot coating (Kärcher, 1998; Yu et al., 1999; Kärcher et al., 2007), but does not occur in our ground setup due to the absence of nucleation mode particles. Thus, soot particles in this study are expected to have a lower amount of coating from this effect. On the other hand, total particle surface area was less in our ground setup due to the absence of the nucleation mode particles, and the exhaust gases experience reduced dilution in the aerosol reservoir with synthetic air, compared to the strong dilution that would occur within the first seconds at flight altitudes (Karcher et al., 2007). This effect would enhance the condensation of volatiles onto the soot particles in our ground setup. However, even if lower amounts of organics and sulfate condense onto the soot particles in the atmosphere, these would first condense into the pores of the soot, due to the capillary effect, and thus inhibit ice nucleation of the soot particles. Thus, our conclusions of the poor ice nucleating ability of contrail processed soot would remain the same. Downstream of HINC1, the reemitted soot aggregates get compacted due to contrail processing and their sizes decrease. The formation of large soot aggregates (Petzold et al., 1998, 1999) due to the coagulation between ice crystals and scavenging of interstitial soot aggregates was not possible due to the low concentration of ice crystals and soot and non-turbulent flow in HINC1. Due to the absence of nucleation

mode particles, coagulation of these with ice crystals, and coagulation with interstitial soot aggregates was not possible in HINC1. However, the absence of these processes does not change our conclusions as adding more organics onto the soot particles would only further result in poor ice nucleation activity (Testa et al. 2024, Gao and Kanji, 2022a). Gas phase chemistry and particle oxidation are thought to considerably slow down while exiting the combustor chamber due to much lower temperatures in the exhaust nozzle and downstream of the engine (Dakhel et al., 2007; Wong et al., 2008). Such a drop in temperature was also present in our sampling system (thousand degrees to 433 K and to room temperature), thus the primary particle overlap, size, crystallinity, and oxidation should be unaffected and comparable to in situ emitted aviation soot particles.

Summarizing, soot particles in our ground setup were larger and less dense than in situ soot due to coagulation in our aerosol reservoir. We believe the aggregate compaction in this work is atmospherically relevant as the parameters driving the soot compaction, i.e., $RH_i$ experienced by the particles and bulk water condensation were represented in our ground setup. The primary particle properties and oxidation are fixed in the combustor and hence should be representative of their in-situ counterpart. Finally, we expect in situ particles to be coated with $H_2SO_4$ and organics but to which extent the coating of the aviation soot sample in our study is different from in situ aviation soot cannot be established from our study. For this reason, we quantified the ice nucleation ability of coating free (CS-CP-soot) and coated (unprocessed and CP-soot) soot in our study to constrain the possible effect of different soot mixing states on aviation soot ice nucleation."

**R1C2: Line 42-43: The gap between the exhaust and the inlet, introducing dilution and turbulence: would it impact the coagulation in the tank? would a higher concentration sampled directly from the engine have resulted in a faster coagulation, a different size mode? if so are you properly simulating the contrail formation process?**

AR2: As mentioned above, the sampling unit meets the ICAO requirements for engine emission measurements (Durdina et al., 2014), hence there is no air intrusion between the exhaust nozzle and the sampling probe. As explained in the paper, the sampled soot particles were confined in the restricted volume of the aerosol reservoir for several hours, where they coagulated causing a shift in the particle size distribution to larger sizes and less dense particles. We agree, the soot emission index of the different engines (soot number concentration measured at the engine exit plane) impacts the coagulation occurring in the aerosol reservoir with larger soot emission indices resulting in faster and more efficient coagulation, hence larger soot aggregate sizes. For this reason, the soot mode diameters were different for each engine test and ranged between 80-450 nm (as stated in line (47) 54).

The size of the soot aggregate could impact the activation properties to contrail ice crystals as larger particles might easily activate as cloud droplets. Yet, AF values measured at contrail conditions in HINC1 (20 to 100 %) are in the range of what is expected for contrail formation (Kärcher et al., 2015). The size of the soot aggregates would mainly affect their ability to nucleate ice via PCF (e.g., Mahrt et al., 2018; Marcolli et al., 2021; Zhang et al., 2020) rather than via homogeneous freezing hence affecting their ability to perturb natural cirrus rather than to form contrail (whose determining properties is number concentration; Kärcher et al., 2018). The effect of size, discussed in Testa et al. (2024) and in the present paper (lines (235-243) 294-302), strengthen the conclusion that the non-contrail processed, and contrail processed aviation soot would nucleate ice at or above $RH_{hom}$ and only a small fraction (below 0.1 %) nucleates ice below $RH_{hom}$.

**R1C3: The aforementioned meter gap between the engine exhaust and the sampling inlet: how injection of ambient air downstream of jet fuel combustion would impact soot emissions, concentrations, oxidation etc. I would like to see a discussion referring to findings reported by Kelesidis et al. 2023.**

AR3: As mentioned above, there is no injection of air in the meter gap between the exhaust nozzle and the sampling probe. The only injection of (synthetic) air occurred in the aerosol reservoir during the ice nucleation measurements. In opposition to the study by Kelesidis et al. (2023) who injected $O_2/N_2$ in the combustor chamber (where the gas temperature is high), the low temperature of the air in our setup (ambient temperature) would not allow oxidation of the soot surface and condensed organics (Raj et al., 2013, 2014).

**R1C4: What was the temperature of air intake by the engine? Would room temperature air intake affect the combustion products? The characteristics of soot particles formed? How different would it be in comparison to intake of -60C cold air?**

AR4: The temperature of the air flow in the test cell and hence entering the engine was similar to the (outside) ambient temperature. The main parameter affecting the performance of aircraft engines is the air density, with lower density leading to lower engine thrust (Balicki et al., 2014). The air density at ground is higher than at flight altitudes resulting in higher thrust at ground (1.2 kg/m$^{-3}$ at ground and 0.3 kg/m$^{-3}$ at 200 hPa and - 60°C, i.e., flight altitude). The effect of engine thrust on particle properties and ice nucleation was investigated and discussed in Testa et al. (2024).

**R1C5: Given the fact that the engines tested had differences between the produced soot, can the results of this study be generalized to the extent expressed in the title?**

AR5: We agree that differences in soot chemical properties, such as functional groups and sulfur content, and soot morphology, were observed in between the different engines tested (reported in Testa et al., 2024 and in this study). The more ice active soot sampled in our study (CP-CS-soot sample) show very modest ice nucleation ability (compared to aviation soot proxies), with a minimum $RH_i$ onset observed at 143 % (at -55°C). Yet, aviation soot that were contrail processed only (CP-soot sample) are thought to be more representative of flight altitude aviation soot (lines (232-234) 291-293) and their ice nucleation ability, for all different engines tested, converge to $RH_{hom}$ (Figure 2). Lower $RH_i$ onset at 137 % and 140 % were observed in Testa et al. (2024) but only for large (> 400 nm) and hence unrealistic aggregate sizes. As emphasized in the paper, the number and size of cavities within the soot aggregates are the primary controlling factors of soot ice nucleation via PCF. This is primarily controlled by the primary particle morphology and the aggregates size. High overlap of the primary particles has been observed on soot samples for all tested engines. Smaller aggregate size for aviation soot is expected for turbofan engines, hence we believe that the results from our study, that is, aviation soot particles are poor ice nucleating particle for cirrus formation can be generalized to soot emitted with the current aircraft fleet and fuel (jet A/A-1), as emphasized in the title because the size is of the particles is too small to nucleate ice other than by homogeneous freezing of droplets.

Lines 284-293 now read: "Aviation soot samples that were catalytically stripped and contrail processed were able to nucleate ice around 145 % $RH_i$ at 218 K (~7 % lower than $RH_{hom}$). The modest ice nucleation ability for the CS-CP-soot likely arises from increased cavity number and sizes within the soot aggregates, which would be absent in the unprocessed soot samples coated with organics and sulfate. […] Thicker coatings for flight altitude aviation soot compared to our CP-soot would favor homogeneous nucleation and thinner coatings would be bound by the ice nucleation ability of our CS-CP-soot sample. Nevertheless,

as long as aviation soot is co-emitted with $H_2SO_4$, it is likely to acquire a coating upon emission and further in the exhaust plume (Kärcher et al., 2007), thus we expect our unprocessed and CP-soot samples to be of higher atmospheric relevance for engines and fuel currently in use."

To complete the soot size argument in lines (235-243) 294-302, we proposed to add the following (lines 303-309 in the revised manuscript): "The number and size of cavities within the soot aggregates are the primary controlling factors of soot ice nucleation via PCF (Sect. 1). The cavity formation is controlled by the primary particle morphology (being determined in the engine combustor and therefore well simulated in our ground setup) and the aggregates size. High overlap of the primary particles has been observed on soot samples for all tested engines. Smaller aggregate size for aviation soot is expected for turbofan engines, hence we believe that the results from our study, that is, aviation soot particles are poor ice nucleating particle for cirrus formation can be generalized to soot emitted with the current aircraft fleet and fuel (Jet A/A-1; > 90 % of global usage; Jing et al., 2022; Pires et al., 2018)"

**R1C6: Sulfuric acid was mentioned as one of the main factors limiting the PCF. This is more obvious in a long and slow cooling process that allows for more efficient condensation. What about an instant temperature and significant instant pressure drop between the combustor and the high-altitude environment, would that impact the uniformity of sulfuric acid vapors condensation? The boiling point of sulfuric acid is 166 C, would thermodynamics and cooling rate in your experiment, dictate the condensation of sulfuric acid and impact its degree of soot surface coating, which in turn impacts the reactivation capacity due to the chemical content present in the voids?**

AR6: We agree, the ground set up would mainly affect the particle mixing state, i.e., the level of soot surface coating (for instance with $H_2SO_4$), and size, and hence their ice nucleation ability via PCF. For this reason, we performed experiments for the extreme case of catalytically stripped particles (CS-CP-soot sample) by removing the organics and sulfur coated on the soot with a catalytic stripper. The ice nucleation ability of the sulfur free aviation soot was better than the coated soot, but still remains poor (see Fig 2). We believe that PCF for our catalytically stripped soot sample is limited by the availability and morphology of inter-aggregate cavities, primarily governed by the primary particle morphology, which is not affected by our setup and sampling method but rather fixed in the engine combustor (see AR1.2 and AR5 for proposed text edits).

**R1C7.1: The dynamics, rates of processes are extremely important, dropping the particle and air temperature from thousands of degrees, to hundreds and then to subzero temperature during tens of minutes to several hours is not the same as an almost instantaneous transition from thousands of degrees to -60 C, would soot particle temperature be different than its carrying air temperature, would heat dissipation be slower at low pressure, would that impact INP activity and how would that compare to this slow ground test?**

AR7.1: As mentioned in AR1.2, we acknowledge that the rates of processes are different to in situ rates in our ground-based setup and conclude that this will affect the soot size and mixing state, namely the coating of the sampled soot particles cannot be strictly compared to that of in situ emitted aviation soot. Yet, although in situ measurement of soot mixing state are lacking, we can make the fair assumption that aviation soot would likely acquire a coating during the processing of the aircraft exhaust (Kärcher, 1999) for current engine fleet and fuel. Thus, we quantified the ice nucleation abilities of coating free (CS-CP-soot) and coated (unprocessed and CP-soot) soot in our study to bound the possible effect of different aviation soot mixing state. A larger or smaller soot coating compared to our CP-soot sample would favor homogeneous nucleation or be bound by our CS-CP-soot sample, respectively. Next, as mentioned above,

the ice nucleation is controlled by the limited availability and morphology of inter-aggregate cavities, primarily governed by the primary particle morphology, which is not affected by our setup and sampling method but rather fixed in the engine combustor.

**R1C7.2: Moreover, with the isokinetic flow into the cloud chamber slowly transitioning from room temperature around the inlet of CFDC into -60 C, would you expect glass transition of some of the coatings at higher cooling rates in the atmosphere, would that impact the INP activation? good to discuss the differences.**

AR7.2: The reviewer is correct in raising a valid point that the cooling from room temperature to the chamber temperature could promote glassy aerosol coatings on the soot particles. This would depend on if a contrail forms in the atmosphere or not immediately after the exhaust plume forms. If a contrail forms, water condenses onto the soot particles immediately upon exhaust emission into the atmosphere and then freezes to form a contrail. In this case the glass aerosol formation is not favored because of the high humidity (Kilchhofer et al., 2021). As such this would not be relevant for contrail processing. However, in case a contrail does not form, glassy aerosol coatings could form, and may promote ice nucleation via deposition (Knopf et al., 2018) but for a small AF. Cooling of the soot particles to -60°C is instantaneous once the particles enter HINC (as for in situ emitted soot), i.e., glass transition of condensed organics can occur in the atmosphere and could have occurred in our study. If glassy transitions were occurring in our experiments, this would also mean the soot particles should be more effective INPs than what we observe in our studies, but our data do not support the formation of glassy coatings in our experiments since we observe very low to insignificant activated fractions (unprocessed and CP-soot nucleate ice above $RH_{hom}$).

Lines 287-289 now read: "We note that, if glassy organic coatings formed on the contrail processed (and unprocessed) soot particles, these particles may form ice by deposition nucleation (Knopf et al., 2018), which was not observed due to the high ice nucleation onset above $RH_{hom}$"

**R1C8: The importance of surface properties is discussed e.g., hydrophilic surface oxygenated functionalities, polar groups etc. What's the impact on surface properties of soot aerosol under increased ozone and UV-b, UV-a conditions once emitted at high altitude in comparison to this ground experiment? would proper simulation with exposure to ozone and UV-b would change the results of this experiment? After contrail sublimation multiple things can happen to the soot particles, depending on the timespan, but even in the case of almost immediate reactivation, would ozone and UV change surface properties and the results of the next activation cycle?**

AR8: Oxidation of aerosols by $O_3$ and OH radical can occur in the upper troposphere and has been investigated for several soot types in past studies (Browne et al., 2015; Dymarska et al., 2006; Friebel et al., 2019; Friebel & Mensah, 2019; Gao & Kanji, 2024; Ghio et al., 2020; Han et al., 2016). The combined results of these studies show that the exposure of $O_3$ or OH to soot of different sizes and with different organic fractions, impacts the soot properties differently. For highly graphitic soot with low organic fraction, exposure to ozone increases the soot surface water uptake capacity due to increased hydrophilic groups. For organic-rich soot, several processes might occur. Condensed organics might desorb due to the breaking of covalent bounds with the elemental soot fraction. The oxidized organic products can recondense onto the soot due to their lowered volatility, potentially filling the soot aggregate cavities. Furthermore, oxidized organics, e.g., short alkene and aldehyde that are soluble in acidic solution (e.g., $H_2SO_4$ solution; Yu et al., 1999) can participate to freezing point depression if condensed in pores or they can prevent water uptake by blocking the pores if there are hydrophobic, inhibiting PCF (Gao & Kanji, 2024). Overall if organics are present on the soot particles, ozone oxidation will not promote ice nucleation to conditions below $RH_{hom}$.

In addition, even if organics are absent on the soot particles, the increase in hydrophilicity from ozone adsorption is still secondary for ice nucleation by PCF, and the abundance of pores is still the primary controlling factor (Gao & Kanji, 2024). Since the aviation soot particles lack pores relevant for PCF and possess an organic coating, the aging in the presence of ozone or OH and UV will not enhance their ice nucleation activity.

We expect aviation soot to be coated with organics (PAH, smaller hydrocarbons, aliphatic; Abegglen et al., 2016; Marhaba et al., 2019; Parent et al., 2016) and thus following the above observations from the literature, we can say that the exposure of (organic-rich) aviation soot to $O_3$ or OH at flight altitude would decrease their ability to nucleate ice via PCF due to pore blocking and/or freezing point depression (with oxidation on the time scale of hours; Friebel & Mensah, 2019). An eventual increase in soot water uptake capacity from $O_3$/OH oxidation and thus an enhancement of aviation soot ice nucleation via PCF would remain limited owing to the lack of aggregate mesopores observe for aviation soot in this study and in the companion study. Nevertheless, we do not rule out that oxidation of soot surface organics can also turn into glass and promote heterogeneous ice nucleation of the particles (Tian et al., 2022).

Lines 310-318 now read: "In addition to contrail processing, several soot aging processes can occur in the atmosphere, such as interaction with background aerosols and volatile compounds, or oxidation of aerosols by $O_3$ and OH radicals (Bond et al., 2013). Interception of soluble aerosol onto the soot surface would increase the amount of soot coating preventing PCF. Exposure of aviation soot to $O_3$ or OH at flight altitude can cause condensed organics to desorb due to the breaking of covalent bonds with the elemental soot fraction. The oxidized organics could recondense onto the soot due to their lowered volatility, e.g., short alkanes and aldehyde that are soluble in acidic solution (e.g., $H_2SO_4$ solution; Yu et al., 1999) and could lead to a freezing point depression if condensed in pores or prevent water uptake by blocking the pores if there are hydrophobic, inhibiting PCF (Gao & Kanji, 2024). Nevertheless, we do not rule out that oxidation of soot surface organics can also turn into glassy coatings and promote deposition ice nucleation of the particles (Tian et al., 2022)."

Following this edit and edits from AR1.2, AR5 and AR7.2, we move down the comparison of aviation soot ice nucleation with mineral dust that was out of place (now at lines 319-321): "Additionally, the presence of other potent atmospheric INPs would further limit the effect of aviation soot on cirrus cloud microphysical properties. For instance, we note that the CS-CP-soot $RH_i$ onset remains substantially above that for mineral desert dust, e.g., about 120 % (Ullrich et al., 2017), which all outcompete soot ice nucleation."

**R1C9: On the same topic of surfaces charges, which seem to play a role in the ice nucleation process. Charging is often observed in expansion chambers experiments (e.g., AIDA). I haven't seen any discussion about charging and charge transfer of soot in the exhaust ejection process, would it play a role in measured INP activity in comparison to long mixing in the tank on the ground and isokinetic low flow aspiration with HINC?**

AR9: Charging effects were reported to be significant for soot cluster coagulation within the combustor (high number of charges per particle due to the high temperature) but negligible once the particles exit the combustor (only slightly positively charged [+1 in average]; Dakhel et al. 2007). Charging effect would nonetheless impact the formation of nucleation mode particles for in situ jet plume (Kärcher et al., 2015) but these were curbed in our setup.

**R1C10: Is there any impact of exhaust inhomogeneity and selected sampling inlet location/orientation?**

AR10: The sampling unit was designed to meet the ICAO requirements for engine emission measurements, therefore we do not expect sample inhomogeneity induced by the inlet of the sampling unit.

**R1C11: Does this setup represent well the impact of shear forces trapping the exhaust in vortices behind the aircraft, would this impact the results? could the vortices create multiple sharp supersaturation and sublimation cycles? How does that compare to the timespan of this experiment?**

AR11: We agree with the comment, aircraft exhaust can be trapped in the vortices forming in the wake of the aircraft, where entrainment of air and vertical ascent/descent of the plume can create large supersaturation and freezing/sublimation cycle of the soot particle (Kärcher, 2018; Kärcher et al., 2015; Miake-Lye et al., 1993). We believe that coagulation of the soot particles (and hence effect on their sizes) would remain limited as measured with recent in situ study (Moore et al., 2017). Soot shape and mixing state could nonetheless be affected by the freezing/sublimation cycle and interactions with the numerous volatile particles, respectively.

In our ground setup, aircraft wing vortices were not represented as the aircraft engines are tested in isolation from the aircraft frames (and wings) in the test cell. Freezing/sublimation cycle in wing vortices would be essentially similar to contrail processing, as simulated in our study. The impact on aviation soot mixing state was not represented in our setup, but the effect on the soot ice nucleation properties is similar as discussed above (AR1.2 and AR5).

**R1C12: A key achievement in this study (e.g., lines 31-33,224-225 and elsewhere) is the production of jet engine soot aerosol that unlike previously studied BC proxies is more representative of the contrail ice nucleation followed by "contrail" sublimation and reactivation. While the combusted jet fuel better represents the real content of the exhaust, anything downstream the exhaust is still significantly different on many dimensions in comparison to the physics and thermodynamics of high altitude. The author's claim of a greater relevance needs to be further substantiated, alternatively the claims should be toned down (as should be the accompanying paper) on its relevance to actual contrails or discuss in greater breadth the limitations of this study. These questions have to be addressed before the second step of contrail processing can be properly evaluated.**

AR12: We agree that some processes occurring downstream of the aircraft engine are not represented or absent in our ground-based setup and that would affect the aviation soot properties. Those are now discussed in the above and text have been added as indicated in the previous comments.

Next, as mentioned in AR7.1, our CS-CP-soot sample could represent the best possible ice nucleation ability for aviation soot. The fact that in situ emitted soot aggregates have much smaller sizes further strengthen the claim that the large CS-CP-soot sampled in this study represent to maximal ice nucleation potential for aviation soot and that aviation soot would most likely not perturb cirrus cloud due to their poor ice nucleation ability.

**R1C13: Was the CATZ time loscale representative of contrail sublimation rate? Do you think liquid/solid phase chemistry could play a role here, in the context of sulfuric acid dilution?**

AR13: The ice crystals sublimation/growth rate depends on the relative humidity experienced by the ice crystals (Lohmann et al., 2016). $RH_i$ in CATZ is about 60 %, i.e., representative of most upper tropospheric clear sky humidities (Krämer et al., 2020) hence likely encountered by sublimating contrails. We note that compaction of aviation soot aggregates might be hampered if coated with viscous material (e.g., organics)

but that it would nonetheless occur while the soot coating dilutes with liquid water during contrail cloud droplet activation (Corbin et al., 2023).

**R1C14: The paragraphs and sentences dedicated to hypotheses about alternative jet fuel activation e.g., in the atmospheric implications section. In my opinion, these should be avoided. There is no need to hypothesize about experiments that have not been done, especially given the questions remaining about the applicability of the presented ground test results to real contrails. I'd recommend focusing on the applicability and substantiation of the results of this study with the fuels and engines actually tested.**

AR14: We agree that a discussion on the limitation of the study and the implication for aviation soot ice nucleation need to be further developed in our paper. Such a discussion has now been added into the revised version of the paper as part of responses AR1.2, AR5 and AR7.2. We keep the discussion on the size effect, since there is a wealth of literature supporting the dependence of ice nucleation on soot particle size. We emphasize here that small soot size limits PCF will still hold for soot emitted from alternative aviation fuel s (AAFs) as those are smaller or as small of Jet A/A-1 soot (e.g., Moore et al., 2017; Durdina et al., 2021). Besides, the few studies (see references in Sect. 5) investigating other aviation soot properties such as primary particle size and hydrophilicity, point out to no significant changes of those properties and consequently, limited potential for ice nucleation enhancement for AAF soot compared to standard fuel soot. Such discussion is needed to set the priority in future research on the climate impact of aviation.

**Minor comments:**

**R1C15: "Contrail processed…." – the title is misleading, the ground/laboratory experiment doesn't cover the full complexity of the contrail formation process at high altitude nor does it simulate the transition rates in a "violent" contrail formation but rather presents a cloud formation in a steady state environment, feeding well stirred combusted soot at room temperature into an ice nucleation chamber. Please come up with a title that properly describes the ground simulation that you present.**

AR15: We agree with the reviewer comment and change the title to "Simulated contrail processed […]".

**R1C16: Figure 1 caption: "incomplete combustion… emits" - combustion produces, or aircraft emits**

AR16: Lines 1-2 in caption Figure 1 now reads: "From left to right: Due to the incomplete combustion of aviation fuel, aircraft engines emit soot particles […]"

**R1C17: Line 14: "and they exhibit" – they hint on high likelihood of poor ice nucleation ability at cirrus relevant….**

AR17: Lines (14) 14 now reads: "[…] for the first time, and suggest a high likelihood of poor ice nucleation ability at cirrus relevant temperatures […]"

**R1C18: Line 28: the accompanying study (not published) states 2-50 nm. Which one is correct?**

AR18: By definition, the mesopore range covers 2-50 nm pore diameters. Marcolli et al. (2021) show that the 2-30 nm range is relevant for PCF, while pore diameters above 30 nm are most likely too large to get filled with water below water saturation.

To clarify this, we propose to modify lines (28) 28-29: "Pore of relevant diameters for PCF are in the range of 2-30 nm (Marcolli et al., 2021) and fall into the mesopore size range of 2-50 nm (Haul, 1982)."

**R1C19: Figure 1 caption: "forming a contrail cloud due to high concentration of water vapor and cold temperatures" – in the aerosol reservoir depicted, aren't you removing water vapor down to less than 10%? How does that correspond to the process in the atmosphere?**

AR19: In our setup, contrail formation (or ice crystal formation onto the soot particles at contrail conditions) is mimicked in HINC1, there the $RH_w$ is 105 % (see lines (47-59) 54-66).

**R1C20: Line 151-152, similarly compacted regardless of their size: would a higher sublimation rate cause a greater compaction or was the maximum compaction reached?**

AR20: The driver for soot compaction is the coating mechanism, i.e., the rate of condensational growth into the soot capillaries and the amount of condensing material, with the compaction reduced for $RH_w$ > 120 % (less time for capillary formation; Corbin et al., 2023). In our study, contrail processed soot were exposed to water vapor at $RH_w$ = 105 %, i.e., in the range of what required for maximal compaction according to the review study from Corbin et al. (2023). The aspect is discussed in the proposed text edits in AR1.2.

**1C21: Same for line 159, was maximum compaction reached?**

AR21: The convexity/size change of the contrail processed aviation soot sample from this study is very similar to what measured for other cloud/contrail processed soot types (China et al., 2015; Ma et al., 2013; Mahrt et al., 2020). This is explained by the harsh activation process occurring in HINC1 and CATZ (activation in cloud droplets and ice crystals followed by sublimation). Although large compaction was observed in our study (convexity approaching 1), we do not rule out that smaller condensation/sublimation rate in the atmosphere could lead to larger compaction of the particles.

We propose to clarify this as well in the paper. Lines 259-265 now read: "Downstream of HINC1, the reemitted soot aggregates get compacted due to contrail processing and their sizes decrease. We note that, the convexity/size change of the contrail processed aviation soot sample from this study is very similar to what measured for other cloud/contrail processed soot types (Mahrt et al., 2020; China et al., 2015, Ma et al. 2013). This is explained by the harsh activation process occurring in HINC1 and CATZ (activation in cloud droplets and ice crystals followed by sublimation; Corbin et al., 2023). Although large compaction was observed in our study (convexity approaching 1), we do not rule out that smaller condensation/sublimation rate in the atmosphere could lead to larger compaction of the particles. The formation […]".

**R1C22: Figure 5: a colorbar should be included in the main plot and x axis titles on the adjacent box plots.**

AR22: We agree, Figure 5 has been updated to include the color bar.

**R1C23: Line 223 unlikely to promote ice nucleation…**

AR23: We agree, line (223) 236 now reads: "showed that aviation soot is unlikely to promote ice nucleation"

**R1C24: Line245-259, see major comment 14, this doesn't fit into atmospheric implications of the current study. It looks more like a discussion paragraph about future work that can be done. I would personally prefer to see a future work discussion that will present design insights from this study that can be modified to improve the simulation or guidelines to build a future ground facility for more accurate simulations of contrail formation. Alternatively, in this section there could be more discussion about the**

**impact of the results on radiative transfer (depicted in Figure 1), what is the difference in radiative transfer between what was previously assumed and with the new findings.**

AR24: A detailed discussion on the radiation transfer is beyond the scope of this study. With regard to the ice nucleation ability measured for the aviation soot sampled in this study (and in the companion study), together with the limitations of our ground setup and implications for aviation soot ice nucleation properties (discussed in AR1.2-AR11), we consider that our conclusion, e.g. lines (278-279) 355-356 "aviation soot particles would likely not serve as INP for cirrus formation and that current radiative forcing estimates", is justified.

The present study focusses on aerosol properties and ice nucleation. This justifies a discussion on aerosol properties from AAF. Implications for radiative transfer are beyond the scope of the paper, but highly relevant to the topic of aviation soot-cirrus interaction and therefore will be investigated in follow up modelling study.

**R1C25: Line 268 strong compaction (65% convexity increase) – is that the max compaction that could be achieved? What would be the typical difference between a strong and a weak compaction?**

AR25: The measurements show that the initial size impacts the compaction level, with larger compaction for larger aggregates. Dedicated measurements, e.g., as a function of coating level, aggregate sizes and exposed humidities during condensation and sublimation of liquid water, would be needed to assess the maximal reachable compaction. Nonetheless, from the TEM images, the contrail processed aggregates become round shaped with no or few open aggregate branches, as visible in comparison for the unprocessed soot (see Fig. A2). Weak compaction would be characterized by little change of the shape and size of the aggregates, i.e., low convexity and open-branched aggregates.

**R1C26: Line 271: In real contrails, would this condensation be impacted by the rate of the process, would the phase of the organic matter be impacted by the rate of cooling e.g. Zhang et al. 2019. Would that in turn influence the INP activity?**

AR26: As mentioned in AR7.2, organics could have turned glassy due to the fast cooling of the particles once entering HINC1 (as for in situ emitted aviation soot while exiting the exhaust nozzle). Yet, no heterogeneous ice nucleation was observed for the coated, i.e., unprocessed and CP-soot soot samples that could be due to glassy organic phase on the soot surface.

**References:**

**Korhonen, K., Kristensen, T. B., Falk, J., Malmborg, V. B., Eriksson, A., Gren, L., Novakovic, M., Shamun, S., Karjalainen, P., Markkula, L., Pagels, J., Svenningsson, B., Tunér, M., Komppula, M., Laaksonen, A., and Virtanen, A.: Particle emissions from a modern heavy-duty diesel engine as ice nuclei in immersion freezing mode: a laboratory study on fossil and renewable fuels, Atmos. Chem. Phys., 22, 1615–1631, https://doi.org/10.5194/acp-22-1615-2022, 2022.**

**Crawford, I., Möhler, O., Schnaiter, M., Saathoff, H., Liu, D., McMeeking, G., Linke, C., Flynn, M., Bower, K. N., Connolly, P. J., Gallagher, M. W., and Coe, H.: Studies of propane flame soot acting as heterogeneous ice nuclei in conjunction with single particle soot photometer measurements, Atmos. Chem. Phys., 11, 9549–9561, https://doi.org/10.5194/acp-11-9549-2011, 2011.**

**Georgios A. Kelesidis, Amogh Nagarkar, Una Trivanovic, and Sotiris E. Pratsinis, Environmental Science & Technology 2023 57 (28), 10276-10283, DOI: 10.1021/acs.est.3c01048**

**Yue Zhang, Leonid Nichman, Peyton Spencer, Jason I. Jung, Andrew Lee, Brian K. Heffernan, Avram Gold, Zhenfa Zhang, Yuzhi Chen, Manjula R. Canagaratna, John T. Jayne, Douglas R. Worsnop, Timothy B. Onasch, Jason D. Surratt, David Chandler, Paul Davidovits, and Charles E. Kolb, Environmental Science & Technology 2019 53 (21), 12366-12378, DOI: 10.1021/acs.est.9b03317**

References (from author responses):

Abegglen, M., Brem, B. T., Ellenrieder, M., Durdina, L., Rindlisbacher, T., Wang, J., Lohmann, U., & Sierau, B. (2016). Chemical characterization of freshly emitted particulate matter from aircraft exhaust using single particle mass spectrometry. *Atmospheric Environment*, *134*, 181–197. https://doi.org/https://doi.org/10.1016/j.atmosenv.2016.03.051

Balicki, W., Głowacki, P., Szczeciński, S., Chachurski, R., & Szczeciński, J. (2014). Effect of the Atmosphere on the Performances of Aviation Turbine Engines. *Acta Mechanica et Automatica*, *8*, 70–73. https://api.semanticscholar.org/CorpusID:54978680

Bond, T. C., Doherty, S. J., Fahey, D. W., Forster, P. M., Berntsen, T., DeAngelo, B. J., Flanner, M. G., Ghan, S., Kärcher, B., Koch, D., Kinne, S., Kondo, Y., Quinn, P. K., Sarofim, M. C., Schultz, M. G., Schulz, M., Venkataraman, C., Zhang, H., Zhang, S., … Zender, C. S. (2013). Bounding the role of black carbon in the climate system: A scientific assessment. *Journal of Geophysical Research: Atmospheres*, *118*(11), 5380–5552. https://doi.org/https://doi.org/10.1002/jgrd.50171

Browne, E. C., Franklin, J. P., Canagaratna, M. R., Massoli, P., Kirchstetter, T. W., Worsnop, D. R., Wilson, K. R., & Kroll, J. H. (2015). Changes to the Chemical Composition of Soot from Heterogeneous Oxidation Reactions. *The Journal of Physical Chemistry A*, *119*(7), 1154–1163. https://doi.org/10.1021/jp511507d

China, S., Kulkarni, G., Scarnato, B. V, Sharma, N., Pekour, M., Shilling, J. E., Wilson, J., Zelenyuk, A., Chand, D., Liu, S., Aiken, A. C., Dubey, M., Laskin, A., Zaveri, R. A., & Mazzoleni, C. (2015). Morphology of diesel soot residuals from supercooled water droplets and ice crystals: implications for optical properties. *Environmental Research Letters*, *10*(11), 114010. https://doi.org/10.1088/1748-9326/10/11/114010

Corbin, J. C., Modini, R. L., & Gysel-Beer, M. (2023). Mechanisms of soot-aggregate restructuring and compaction. *Aerosol Science and Technology*, *57*, 89–111. https://api.semanticscholar.org/CorpusID:253154808

Dakhel, P. M., Lukachko, S. P., Waitz, I. A., Miake-Lye, R. C., & Brown, R. C. (2007). *Post-Combustion Evolution of Soot Properties in an Aircraft Engine*. https://api.semanticscholar.org/CorpusID:136958147

Durdina, L., Brem, B. T., Abegglen, M., Lobo, P., Rindlisbacher, T., Thomson, K. A., Smallwood, G. J., Hagen, D. E., Sierau, B., & Wang, J. (2014). Determination of PM mass emissions from an aircraft turbine engine using particle effective density. *Atmospheric Environment*, *99*, 500–507. https://doi.org/https://doi.org/10.1016/j.atmosenv.2014.10.018

Durdina, L., Brem, B. T., Elser, M., Schönenberger, D., Siegerist, F., & Anet, J. G. (2021). Reduction of Nonvolatile Particulate Matter Emissions of a Commercial Turbofan Engine at the Ground Level from

the Use of a Sustainable Aviation Fuel Blend. *Environmental Science & Technology*, *55*(21), 14576–14585. https://doi.org/10.1021/acs.est.1c04744

Dymarska, M., Murray, B. J., Sun, L., Eastwood, M. L., Knopf, D. A., & Bertram, A. K. (2006). Deposition ice nucleation on soot at temperatures relevant for the lower troposphere. *Journal of Geophysical Research: Atmospheres*, *111*(D4). https://doi.org/https://doi.org/10.1029/2005JD006627

Friebel, F., Lobo, P., Neubauer, D., Lohmann, U., van Dusseldorp, S., Mühlhofer, E., & Mensah, A. A. (2019). Impact of isolated atmospheric aging processes on the cloud condensation nuclei activation of soot particles. *Atmospheric Chemistry and Physics*, *19*(24), 15545–15567. https://doi.org/10.5194/acp-19-15545-2019

Friebel, F., & Mensah, A. A. (2019). Ozone Concentration versus Temperature: Atmospheric Aging of Soot Particles. *Langmuir*, *35*(45), 14437–14450. https://doi.org/10.1021/acs.langmuir.9b02372

Gao, K., Friebel, F., Zhou, C.-W., & Kanji, Z. A. (2022). Enhanced soot particle ice nucleation ability induced by aggregate  compaction and densification. *Atmospheric Chemistry and Physics*, *22*(7), 4985–5016. https://doi.org/10.5194/acp-22-4985-2022

Gao, K., & Kanji, Z. A. (2022). Impacts of Cloud-Processing on Ice Nucleation of Soot Particles Internally Mixed With Sulfate and Organics. *Journal of Geophysical Research: Atmospheres*, *127*(22), e2022JD037146. https://doi.org/https://doi.org/10.1029/2022JD037146

Gao, K., & Kanji, Z. A. (2024). Influence of Lowering Soot-Water Contact Angle on Ice Nucleation of Ozone-Aged Soot. *Geophysical Research Letters*, *51*(7), e2023GL106926. https://doi.org/https://doi.org/10.1029/2023GL106926

Ghio, A. J., Gonzalez, D. H., Paulson, S. E., Soukup, J. M., Dailey, L. A., Madden, M. C., Mahler, B., Elmore, S. A., Schladweiler, M. C., & Kodavanti, U. P. (2020). Ozone Reacts With Carbon Black to Produce a Fulvic Acid-Like Substance and Increase an Inflammatory Effect. *Toxicologic Pathology*, *48*(7), 887–898. https://doi.org/10.1177/0192623320961017

Han, C., Liu, Y., & He, H. (2016). The photoenhanced aging process of soot by the heterogeneous ozonization reaction. *Phys. Chem. Chem. Phys.*, *18*(35), 24401–24407. https://doi.org/10.1039/C6CP03938C

Jing, L., El-Houjeiri, H. M., Monfort, J.-C., Littlefield, J., Al-Qahtani, A., Dixit, Y., Speth, R. L., Brandt, A. R., Masnadi, M. S., MacLean, H. L., Peltier, W., Gordon, D., & Bergerson, J. A. (2022). Understanding variability in petroleum jet fuel life cycle greenhouse gas emissions to inform aviation decarbonization. *Nature Communications*, *13*(1), 7853. https://doi.org/10.1038/s41467-022-35392-1

Kärcher, B. (1998). Physicochemistry of aircraft-generated liquid aerosols, soot, and ice particles: 1. Model description. *Journal of Geophysical Research: Atmospheres*, *103*(D14), 17111–17128. https://doi.org/https://doi.org/10.1029/98JD01044

Kärcher, B. (1999). Aviation-Produced Aerosols and Contrails. *Surveys in Geophysics*, *20*(2), 113–167. https://doi.org/10.1023/A:1006600107117

Kärcher, B. (2018). Formation and radiative forcing of contrail cirrus. *Nature Communications*, *9*(1), 1824. https://doi.org/10.1038/s41467-018-04068-0

Kärcher, B., Burkhardt, U., Bier, A., Bock, L., & Ford, I. J. (2015). The microphysical pathway to contrail formation. *Journal of Geophysical Research: Atmospheres*, *120*(15), 7893–7927. https://doi.org/https://doi.org/10.1002/2015JD023491

Kärcher, B., Möhler, O., DeMott, P. J., Pechtl, S., & Yu, F. (2007). Insights into the role of soot aerosols in cirrus cloud formation. *Atmospheric Chemistry and Physics*, *7*(16), 4203–4227. https://doi.org/10.5194/acp-7-4203-2007

Kilchhofer, K., Mahrt, F., & Kanji, Z. A. (2021). The Role of Cloud Processing for the Ice Nucleating Ability of Organic Aerosol and Coal Fly Ash Particles. *Journal of Geophysical Research: Atmospheres*, *126*(10), e2020JD033338. https://doi.org/https://doi.org/10.1029/2020JD033338

Knopf, D. A., Alpert, P. A., & Wang, B. (2018). The Role of Organic Aerosol in Atmospheric Ice Nucleation: A Review. *ACS Earth and Space Chemistry*, *2*(3), 168–202. https://doi.org/10.1021/acsearthspacechem.7b00120

Krämer, M., Rolf, C., Spelten, N., Afchine, A., Fahey, D., Jensen, E., Khaykin, S., Kuhn, T., Lawson, P., Lykov, A., Pan, L. L., Riese, M., Rollins, A., Stroh, F., Thornberry, T., Wolf, V., Woods, S., Spichtinger, P., Quaas, J., & Sourdeval, O. (2020). A microphysics guide to cirrus – Part 2:  Climatologies of clouds and humidity from observations. *Atmospheric Chemistry and Physics*, *20*(21), 12569–12608. https://doi.org/10.5194/acp-20-12569-2020

Lacher, L., Lohmann, U., Boose, Y., Zipori, A., Herrmann, E., Bukowiecki, N., Steinbacher, M., & Kanji, Z. A. (2017). The Horizontal Ice Nucleation Chamber (HINC): INP measurements at  conditions relevant for mixed-phase clouds at the High Altitude  Research Station Jungfraujoch. *Atmospheric Chemistry and Physics*, *17*(24), 15199–15224. https://doi.org/10.5194/acp-17-15199-2017

Lohmann, U., Lüönd, F., & Mahrt, F. (2016). *An introduction to clouds: From the microscale to climate*. Cambridge University Press.

Ma, X., Zangmeister, C. D., Gigault, J., Mulholland, G. W., & Zachariah, M. R. (2013). Soot aggregate restructuring during water processing. *Journal of Aerosol Science*, *66*, 209–219. https://doi.org/https://doi.org/10.1016/j.jaerosci.2013.08.001

Mahrt, F., Kilchhofer, K., Marcolli, C., Grönquist, P., David, R. O., Rösch, M., Lohmann, U., & Kanji, Z. A. (2020). The Impact of Cloud Processing on the Ice Nucleation Abilities of Soot Particles at Cirrus Temperatures. *Journal of Geophysical Research: Atmospheres*, *125*(3), e2019JD030922. https://doi.org/https://doi.org/10.1029/2019JD030922

Mahrt, F., Marcolli, C., David, R. O., Grönquist, P., Barthazy Meier, E. J., Lohmann, U., & Kanji, Z. A. (2018). Ice nucleation abilities of soot particles determined with the Horizontal Ice Nucleation Chamber. *Atmospheric Chemistry and Physics*, *18*(18), 13363–13392. https://doi.org/10.5194/acp-18-13363-2018

Marcolli, C., Mahrt, F., & Kärcher, B. (2021). Soot PCF: pore condensation and freezing framework for soot aggregates. *Atmospheric Chemistry and Physics*, *21*(10), 7791–7843. https://doi.org/10.5194/acp-21-7791-2021

Marhaba, I., Ferry, D., Laffon, C., Regier, T. Z., Ouf, F.-X., & Parent, P. (2019). Aircraft and MiniCAST soot at the nanoscale. *Combustion and Flame*, *204*, 278–289. https://doi.org/https://doi.org/10.1016/j.combustflame.2019.03.018

Miake-Lye, R. C., Martinez-Sanchez, M., Brown, R. C., & Kolb, C. E. (1993). Plume and wake dynamics, mixing, and chemistry behind a high speed civil transport aircraft. *Journal of Aircraft*, *30*(4), 467–479. https://doi.org/10.2514/3.46368

Möhler, O., Adams, M., Lacher, L., Vogel, F., Nadolny, J., Ullrich, R., Boffo, C., Pfeuffer, T., Hobl, A., Weiß, M., Vepuri, H. S. K., Hiranuma, N., & Murray, B. J. (2021). The Portable Ice Nucleation Experiment (PINE): a new online instrument for laboratory studies and automated long-term field observations of ice-nucleating particles. *Atmospheric Measurement Techniques*, *14*(2), 1143–1166. https://doi.org/10.5194/amt-14-1143-2021

Moore, R. H., Thornhill, K. L., Weinzierl, B., Sauer, D., D'Ascoli, E., Kim, J., Lichtenstern, M., Scheibe, M., Beaton, B., Beyersdorf, A. J., Barrick, J., Bulzan, D., Corr, C. A., Crosbie, E., Jurkat, T., Martin, R., Riddick, D., Shook, M., Slover, G., … Anderson, B. E. (2017). Biofuel blending reduces particle emissions from aircraft engines at cruise conditions. *Nature*, *543*(7645), 411–415. https://doi.org/10.1038/nature21420

Parent, P., Laffon, C., Marhaba, I., Ferry, D., Regier, T. Z., Ortega, I. K., Chazallon, B., Carpentier, Y., & Focsa, C. (2016). Nanoscale characterization of aircraft soot: A high-resolution transmission electron microscopy, Raman spectroscopy, X-ray photoelectron and near-edge X-ray absorption spectroscopy study. *Carbon*, *101*, 86–100. https://doi.org/https://doi.org/10.1016/j.carbon.2016.01.040

Petzold, A., Döpelheuer, A., Brock, C. A., & Schröder F. (1999). In situ observations and model calculations of black carbon emission by aircraft at cruise altitude. *Journal of Geophysical Research: Atmospheres*, *104*(D18), 22171–22181. https://doi.org/https://doi.org/10.1029/1999JD900460

Petzold, A., Ström, J., Ohlsson, S., & Schröder, F. P. (1998). Elemental composition and morphology of ice-crystal residual particles in cirrus clouds and contrails. *Atmospheric Research*, *49*(1), 21–34. https://doi.org/https://doi.org/10.1016/S0169-8095(97)00083-5

Pires, A. P. P., Han, Y., Kramlich, J., & Garcia-Perez, M. (2018). Chemical Composition and Fuel Properties of Alternative Jet Fuels. *Bioresources*, *13*(2), 2632–2657. https://doi.org/10.15376/biores.13.2.2632-2657

Testa, B., Durdina, L., Alpert, P. A., Mahrt, F., Dreimol, C. H., Edebeli, J., Spirig, C., Decker, Z. C. J., Anet, J., & Kanji, Z. A. (2024). Soot aerosols from commercial aviation engines are poor ice-nucleating particles at cirrus cloud temperatures. *Atmospheric Chemistry and Physics*, *24*(7), 4537–4567. https://doi.org/10.5194/acp-24-4537-2024

Tian, P., Liu, D., Bi, K., Huang, M., Wu, Y., Hu, K., Li, R., He, H., Ding, D., Hu, Y., Liu, Q., Zhao, D., Qiu, Y., Kong, S., & Xue, H. (2022). Evidence for Anthropogenic Organic Aerosols Contributing to Ice Nucleation. *Geophysical Research Letters*, *49*(17), e2022GL099990. https://doi.org/https://doi.org/10.1029/2022GL099990

Wong, H.-W., Yelvington, P. E., Timko, M. T., Onasch, T. B., Miake-Lye, R. C., Zhang, J., & Waitz, I. A. (2008). Microphysical Modeling of Ground-Level Aircraft-Emitted Aerosol Formation: Roles of Sulfur-

Containing Species. *Journal of Propulsion and Power*, *24*(3), 590–602. https://doi.org/10.2514/1.32293

Yu, F., Turco, R. P., & Kärcher, B. (1999). The possible role of organics in the formation and evolution of ultrafine aircraft particles. *Journal of Geophysical Research: Atmospheres*, *104*(D4), 4079–4087. https://doi.org/https://doi.org/10.1029/1998JD200062

Zhang, C., Zhang, Y., Wolf, M. J., Nichman, L., Shen, C., Onasch, T. B., Chen, L., & Cziczo, D. J. (2020). The effects of morphology, mobility size, and secondary organic aerosol (SOA) material coating on the ice nucleation activity of black carbon in the cirrus regime. *Atmospheric Chemistry and Physics*, *20*(22), 13957–13984. https://doi.org/10.5194/acp-20-13957-2020

---

## Author Comment (AC2)

Response to egusphere-2024-151 reviews for RC2

**Referee comments are marked in black bold and are numbered as R1Cx with x the comment number.** Author (AC) responses are marked in black directly below the comments. The original text from the manuscript is repeated in blue and corrected text in revised manuscript is typed in red. The former line numbers are given in "()" followed by the new line numbers.

**In this study, Testa et al. investigate ice nucleation abilities of aerosols emitted by six different commercial aircraft engines fueled with Jet A-1 and running while on the ground. The authors further subject the aircraft-engine emitted soot aerosols to catalytic stripping and/or cloud chamber processing to simulate contrail clouds conditions and evaluate their impact on the aerosols' ice nucleation activity at temperatures relevant for cirrus formation. The authors investigate the effects of H2SO4 and volatile organic coating, the effect of morphology and compaction of soot aggregates on their ice nucleation ability. The authors conclude that H2SO4 and volatile organic coatings inhibit ice nucleation in preventing PCF to occur and that the compaction observed after "contrail-like processing" remained inefficient at promoting PCF, which makes aircraft-engine emitted unprocessed and processed soot unlikely INPs.**

**The manuscript is of good quality and the study relevant. However, the manuscript would benefit from a clear assessment of its potential limitations with respect to its representativity of real contrail processing conditions and thus cannot be generalized to the extent expressed in the title. I would therefore recommend this manuscript for publication after this major concern has been addressed, particularly in the experimental section.**

We thank the reviewer and answer to the comments/questions individually below.

**Some of the comments and questions below can help address this concern.**

**R2C1: I fell the title is misleading as the aerosols were flown through experimental chambers to simulate contrail processing but only to some extent since the experimental conditions were still far from what would happen in real conditions (e.g., high-altitude temperature, pressure, background gas or aerosols, versus ground). Maybe change to a humbler title?**

AR1: We agree with the point raised in the comment. As proposed in the response AR15 to RC1, we propose to modify the title for "Simulated contrail processed aviation soot aerosol are poor ice nucleating particles at cirrus temperatures".

**R2C2: Line 40: several engines' models from Pratt&Whitney and CFM international fueled with Jet A-1 were used to produce soot particles. How are these engines representative of the fleet? What about the fuel? Since their ice nucleation response can be different (as shown in this manuscript), it may be relevant to get an idea from the get go. Please specify.**

AR2: The engine type and fuel used in this study are representative of the major part of commercial airliner. P&W and CFM International engines represented 68 % of the aircraft engine fleet in 2020 (52 % for CFM engines and 16 % for P&W engines; FlightGlobal.com, 2021). Jet A (largely used in North America) and jet A-1 (used in the rest of the world) fuel represent >60 % and 30 % of global fuel consumption, respectively (Jing et al., 2022; Pires et al., 2018), with Jet A and Jet A-1 having very close chemical composition (Pires et

al., 2018) and presumably results in aviation soot with similar properties. For the generalization of the results to the extent expressed in the paper and in the title, please see the response AR5 to R1C5 for a detailed argumentation.

Proposed text changes are presented below (line 284-293 in the revised manuscript): "Aviation soot samples that were catalytically stripped and contrail processed were able to nucleate ice around 145 % $RH_i$ at 218 K (~7 % lower than $RH_{hom}$). The modest ice nucleation ability for the CS-CP-soot likely arises from increased cavity number and sizes within the soot aggregates, which would be absent in the unprocessed soot samples coated with organics and sulfate. […] Thicker coatings for flight altitude aviation soot compared to our CP-soot would favor homogeneous nucleation and thinner coatings would be bound by the ice nucleation ability of our CS-CP-soot sample. Nevertheless, as long as aviation soot is co-emitted with $H_2SO_4$, it is likely to acquire a coating upon emission and further in the exhaust plume (Kärcher et al., 2007), thus we expect our unprocessed and CP-soot samples to be of higher atmospheric relevance for engines and fuel currently in use."

And lines 303-309 in the revised manuscript: "The number and size of cavities within the soot aggregates are the primary controlling factors of soot ice nucleation via PCF (Sect. 1). The cavity formation is controlled by the primary particle morphology (being determined in the engine combustor and therefore well simulated in our ground setup) and the aggregates size. High overlap of the primary particles has been observed on soot samples for all tested engines. Smaller aggregate size for aviation soot is expected for turbofan engines, hence we believe that the results from our study, that is, aviation soot particles are poor ice nucleating particle for cirrus formation can be generalized to soot emitted with the current aircraft fleet and fuel (Jet A/A-1; > 90 % of global usage; Jing et al., 2022; Pires, 2018)."

**R2C3: Line 42: In which medium were the aerosols collected 1m downstream the exhaust nozzle, ambient air and room temperature? Do you expect this difference with real conditions to affect aerosols' properties, if yes how so?**

AR3: The engines were fed with air at ambient temperature and humidity. The exhaust was sampled downstream of the engine without injection of air between the exhaust nozzle and the sampling probe, as such, the sample particles are representative of engine exit plane soot particles (response AR2 to R1C2 and AR3 to R1C3). Differences in particle properties could come from the low altitude of soot emission and measurement (ground compared to flight altitude) and are discussed in response AR4 to R1C4.

**R2C4: Line 43: the exhaust is directed to an aerosol reservoir; do you expect any change in the aerosols' mixing state in the tank? Is this directly comparable to what happens in real conditions? What is the temperature in the tank? Are the air composition/energy carriers (e.g. UV photons) simulated the same way as in real conditions? This should be discussed so the reader can understand the potential limitations.**

AR4: We agree with the comment, the exhaust temperature and the particle sampling in the mixing tank in our setup would influence the soot particles physicochemical properties. The limitation of our setup and expected difference in the sampled soot properties compared to what would be expected for in situ aviation soot particles are discussed in the response AR1.2 to R1C1.2.

Regarding the setup description, we propose the following edits. Lines (39-44) 40-51 now read: "The experimental setup was designed to simulate targeted atmospheric processes and to mimic the contrail processing of the sampled aviation soot particles (Fig. 1a). Limitations of the ground setup in representing

atmospheric processes (e.g., aircraft exhaust evolution, contrail formation) and soot ice nucleation ability are discussed in Sect. 5. Soot particles were sampled from in-use commercial aircraft engines (multiple models from Pratt & Whitney and CFM International) running in an indoor test cell, with air intake at ambient temperature and humidity. The engines were all fueled with Jet A-1 fuel and ran from low to high power (30-100 % sea level thrust). The detailed sampling system is described in Testa et al. (2024) which we briefly describe here. The engine exhaust gas and particles were sampled by a heat-resistant alloy probe ~1 m downstream of the engine exhaust nozzle and directed by a long trace-heated line (12 m at 433 K) to a stirred tank, that was in a room next to the engine test cell and acted as an aerosol reservoir where the soot particles accumulated and coagulated (Fig. 1b). The exhaust temperature drops in three steps, from the engine temperature (thousands of Kelvin) to the heated line temperature (433 K) and then to room temperature followed by a third drop from room temperature to the cloud chamber temperature (< 228 K, see below)".

Line (49-50) 57 now reads: "[…] set to contrail cloud thermodynamic conditions ($T$ = 228 K and $RH_w$ = 105 %, HINC1 in Fig. 1b) allowing […]".

We also made the following changes to the text in response to the comment above. The title of Sect. 5 has been modified for: "5 Atmospheric implications and limitations"

Lines (238) 242-283 in Sect. 5 now read: ""[…] leaving cirrus cloud properties essentially unperturbed. Due to constraints of the measurement facility in this work, the representation of aircraft exhaust processes in the ground set up and hence of the aviation soot properties might differ from those at flight altitude. For instance, differences in equilibrium temperatures and dilution would impact the partitioning of volatile unburned hydrocarbons and sulfur compounds and hence the soot ice nucleation ability. The higher temperatures in our setup (433 K, then room temperature, Sect. 2.1) compared to upper tropospheric temperature (< 228 K; Krämer et al., 2020), would not promote the condensation of volatiles onto soot as much as would occur at flight altitude temperatures. The interaction of nucleation mode particles with aviation soot in the young aircraft plume downstream of the engine is thought to increase the soot coating (Kärcher, 1998; Kärcher et al., 2007; Yu et al., 1999), but does not occur in our ground setup due to the absence of nucleation mode particles. Thus, soot particles in this study are expected to have a lower amount of coating from this effect. On the other hand, total particle surface area was less in our ground setup due to the absence of the nucleation mode particles, and the exhaust gases experience reduced dilution in the aerosol reservoir with synthetic air, compared to the strong dilution that would occur within the first seconds at flight altitudes (Karcher et al., 2007). This effect would enhance the condensation of volatiles onto the soot particles in our ground setup. However, even if lower amounts of organics and sulfate condense onto the soot particles in the atmosphere, these would first condense into the pores of the soot, due to the capillary effect, and thus inhibit ice nucleation of the soot particles. Thus, our conclusions of the poor ice nucleating ability of contrail processed soot would remain the same. Downstream of HINC1, the reemitted soot aggregates get compacted due to contrail processing and their sizes decrease. The formation of large soot aggregates (Petzold et al., 1998, 1999) due to the coagulation between ice crystals and scavenging of interstitial soot aggregates was not possible due to the low concentration of ice crystals and soot and non-turbulent flow in HINC1. Due to the absence of nucleation mode particles, coagulation of these with ice crystals, and coagulation with interstitial soot aggregates was not possible in HINC1. However, the absence of these processes does not change our conclusions as adding more organics onto the soot particles would only further result in poor ice nucleation activity (Gao & Kanji, 2022; Testa et al., 2024). Gas phase chemistry and particle oxidation are thought to considerably slow down while exiting the combustor chamber due to much lower temperatures in the exhaust nozzle and

downstream of the engine (Dakhel et al., 2007; Wong et al., 2008). Such a drop in temperature was also present in our sampling system (thousand degrees to 433 K and to room temperature), thus the primary particle overlap, size, crystallinity, and oxidation should be unaffected and comparable to in situ emitted aviation soot particles.

Summarizing, soot particles in our ground setup were larger and less dense than in situ soot due to coagulation in our aerosol reservoir. We believe the aggregate compaction in this work is atmospherically relevant as the parameters driving the soot compaction, i.e., $RH_i$ experienced by the particles and bulk water condensation were represented in our ground setup. The primary particle properties and oxidation are fixed in the combustor and hence should be representative of their in-situ counterpart. Finally, we expect in situ particles to be coated with $H_2SO_4$ and organics but to which extent the coating of the aviation soot sample in our study is different from in situ aviation soot cannot be established from our study. For this reason, we quantified the ice nucleation ability of coating free (CS-CP-soot) and coated (unprocessed and CP-soot) soot in our study to constrain the possible effect of different soot mixing states on aviation soot ice nucleation."

**R2C5: Line 49-51: how long do the aerosols remain in the first cloud chamber? In the subsaturated flow tube? Do these residence times have an influence?**

AR5: The residence times in both cloud chambers and the sublimation flow tube are about 10s each (Testa et al., 2024; Mahrt et al., 2020). Once the soot particles nucleate ice in the first cloud chamber, we do not expect their properties to evolve with time (slow molecular diffusion within the ice phase). In the sublimation flow tube, we could envision that longer residence time could allow the volatile material coating the soot particles to further partition in the gas phase, changing the soot mixing state. The effect of aviation soot mixing state in the ability of aviation soot to reactivate as ice crystal in cirrus have been investigated in this study and in the companion study by catalytically stripping the particles.

**R2C6: Line 64: what is the efficiency of the catalytic stripper, it is not specified?**

AR6: The efficiency of removal of organic volatile compounds is about 99 % but can degrade over time due to sulfur depositing on the catalyst. In the companion paper (Testa et al., 2024), we quantified the soot particle mass loss upon catalytic stripping for different engine type and particle size (we measured up to 10 % mass loss) and we refer to those measurements in the present paper rather than on the catalytic stripper's efficiency specified by the constructor.

**R2C7: Line 70: do you expect any sampling-induced change in morphology when collecting particles on TEM grids? This should be discussed as results interpretations are based on this morphology analysis.**

AR7: We agree with the comment. Soot aggregates can (partially) break apart upon impaction (Gao & Kanji, 2022), especially if the bonds between soot monomers are fragile due to prior heating (as for our CS-soot samples). We however did not detect any soot aggregate fragments with sizes similar to individual primary particles on the TEM images that would indicate aggregate break-up. We also did not observe a morphology difference between the unprocessed and the CS-soot samples, ruling out fragmentation of the heated particles upon impaction. Other possible effects are flattening of the particles, possibly more pronounced for large aggregates or bouncing of the particle on the grid, causing structural change (Virtanen et al., 2010). Such effects are likely limited with the Partector TEM sampler using electrostatic softer (than aerodynamic) impaction.

Second, multiple soot particles might aggregate on the TEM grid if impacting at the same position. This is however limited for with electrostatic impaction with the Partector TEM sampler. Nonetheless, we avoided imaging particles that would be the result of aggregation on the grid upon compaction by carefully imaging clearly isolated soot aggregates.

We propose to add the following edits in lines (73) 83-85: "The Partector TEM sampler uses a soft particle impaction technique (electrostatic precipitation), limiting the effect of the sampling process on the particle morphology. Soot aggregate breaking upon impaction was not observed on the various soot samples. Only clearly isolated soot particles on the grid were imaged to avoid imaging aggregated particles on the grid. Individual soot aggregates were imaged with a JOEL-JEM […]"

**R2C8: Line 106: "trigger modest PCF at 5% RHi […]": Why is this qualifier used? Modest compared to what? The nucleation onset is reached at RHi below RHhom, so it does trigger PCF, doesn't? If it is implicitly compared to other potential INPs, please make it explicit.**

AR8: The qualifier "modest" refers to a comparison with aviation soot proxies (e.g., Marth et al., 2020). To remain clear, we have deleted the word "modest" and kept the explicit comparison with the onset RH values.

Line (106) 118 now reads: "trigger PCF at 5% RHi […]"

**R2C9: Line 112: Were all the engines used for each sample type to derive the results shown in Figure 2? It is not specified.**

AR9: The ice nucleation ability of unprocessed and CP-soot were systematically measured for all engines tested (13). The ice nucleation ability of the CS-soot and CS-CP-soot were measured for only 5 out 13 engines because of time constraints and the fact that we were piggy backing on the engine testing experiments, i.e. the engines were not explicitly run for our measurements. This would be too costly. Figure 2 includes all ice nucleation measurements (the number of engines is indicated in the x-axis, as specified in the figure caption).

Line (66) 73-75 now reads: "[…] and catalytically stripped plus contrail processed soot ("CS-CP-soot"). The ice nucleation ability for the unprocessed and CP-soot samples were systematically measured for all engine tested (13 engines in total). The ice nucleation ability of the CS-soot and CS-CP-soot were measured for 5 engines."

**R2C10: Line 125: "for all engines, the soot aggregate mass increases for given sizes". However, it seems that the error bars displayed in Figure 3b prevent any comparison between CP soot and unprocessed soot as the error bars fully overlap. Same goes for the distinction between 150 nm particles and smaller ones regarding their compaction upon "contrail processing". The interpretations should be moderated accordingly.**

AR10: We agree with this remark and propose to change the text accordingly. Lines (124-125) 136-139 now read: "For the engines shown in Fig. 3a, the soot aggregate mass increases for given sizes, which we explain by particle densification through its compaction, resulting in a higher mass-mobility exponent. For the smaller PW4168A engine soot particles ($D_m$ = 150 nm; Fig. 3b), we also note a mass change upon contrail processing although within the measurement uncertainties."

**R2C11: Line 139-141: here again comparisons are done between processed and unprocessed soot for different engines but the stated results (e.g., ΔDm is larger for CP soot compared to CS CP soot) rely on**

**differences that fall within measurement uncertainties (clearly said this time). More caution should be taken in such case and the interpretations should be moderated accordingly.**

AR11: We agree that differences in size between CP-soot and CS-CP-soot fall within measurement uncertainties. However, the difference is consistent among the three engines (size change for CS-CP-soot always smaller than for CP-soot), and this will be confirmed with the TEM image analysis in Sect. 3.2.3.

We propose the following edits (lines 152-154 in the revised manuscript): "We note that although the difference remains within measurement uncertainties, $\Delta D_m$ is consistently larger for CP-soot compared to CS-CP-soot for all engines (as confirmed in Sect. 3.2.3). This would be expected as soot particles become more hydrophobic upon catalytic stripping, hence less sensitive to compaction upon contrail processing (discussed further in Sect. 4)".

**R2C12: Line 148: "[…] for all investigated engine types, which is indicative of aggregate compaction". What about the previous limitation for particles below 150 nm that showed no compaction in mass measurements; TEM results do not show such limitation?**

AR12: As stated in AR10, we observe a mass change for the PW4168A CP-soot but that falls within the measurement uncertainty. Uncertainties on the mass measurements are larger for the soot particles emitted by the PW4168A engine due to their small particle size (absolute mass change is weaker for lighter particles). The TEM measurements confirm that even the 150 nm particles undergo a morphology change upon contrail processing.

**R2C13: Line 162-163: "CS-CP soot undergo […] than CP soot upon processing". Which processing? Both samples have already been processed. Please clarify/reword.**

AR13: Here we mean "contrail processing". Line 162-163 (175-176) now read: "CS-CP soot undergo […] than the CP-soot".

**R2C14: Line 160-169: "CS-CP-soot show morphology changes similar to CP soot" […] CS-CP-soot undergo smaller size reduction than CP-soot upon processing" […] "CS-CP soot are on average more hydrophobic. This is due to the removal of H2SO4 upon catalytic stripping". I do not understand what parameter is preponderant here to explain why CS-CP-soot need the lowest %RHi to activate ice nucleation compared to CP-soot (Fig 2), is it because of the pore size, hydrophobic character and/or H2SO4 removal? Could you please clarify?**

AR14: As stated in lines (186-188) 199-201 and (197-200) 211-213, the removal of material condensed in the soot pore (e.g., $H_2SO_4$) explain the ability for the CS-CP-soot samples to trigger ice nucleation at lower $RH_i$ compared to the CP-soot sample. Empty pores for the CS-CP-soot become available for water condensation and ice nucleation, while pores partially filled with organics or $H_2SO_4$ for the CP-soot might either not fill with water due to the likely hydrophobic character of the organics or not freeze at the considered temperatures (-45 and -55°C) due to freezing point depression induced by the soluble material in the soot pores.

In addition, the pore size and hydrophobic character of the particles contribute to the ice nucleation ability of the CS-CP-soot and explain differences between the engines (lines (188-195) 201-208).

**R2C15: Line 177: how is "the inability of CP-soot to promote ice nucleation […] contrail processing does not generate pores relevant to PCF" similar to observing moderate enhancement for contrail-processed propane soot. Please remove "similarly" or reword.**

AR15: Line (177) 190 now reads: "[…] mass measurements. Gao and Kanji (2022a) […]"

**R2C16: Line 182: "CFM56-7B", should it be "CFM56-7B26/3" instead?**

AR16: This should be "CFM56-7B26" (same engine as in Testa et al., 2024; see their Table B1). The text has been changed accordingly (line 195 in revised manuscript).

**R2C17: Line 203: possess**

AR17: Text has been edited (see line 216 in revised manuscript).

**R2C18: Line 223: again the very strong statement "aviation soot does not promote ice nucleation below […] RHhom" tends to generalize this to all engines, while in the same paper (Testa et al. 2023) a subset of engines (2/10) were shown to activate ice via PCF when large soot aggregates were emitted. The statement needs to be moderated not to elude this result.**

AR18: In Testa et al. (2024) the size of the ice-active aviation soot particles was large (400 nm monodisperse) and hence not atmospherically relevant. The size of in situ emitted aviation soot are much smaller, which strengthen the statement that aviation soot would not serve as INP for cirrus formation.

To be specific we changed the sentence to (see lines 236-239 in revised manuscript): "Recent measurements (Testa et al., 2023) showed that aviation soot does not promote ice nucleation below conditions required for homogeneous freezing of solution droplets (RHhom), particularly because the sizes of emitted soot particles are below 100 nm, which have been shown to be poor INPs for all conditions (variety of mixing states and particle morphologies)."

This investigation of the effect of the particle mixing state on the ice nucleation with the different soot samples investigated in this study further bound the possible in situ aviation soot ice nucleation ability, hence the generalization of the results. Please refer to AR5, AR6 and AR7.1 in the responses to R1C5, R1C6 and R1C7.1 for a detailed argumentation.

References:

Dakhel, P. M., Lukachko, S. P., Waitz, I. A., Miake-Lye, R. C., & Brown, R. C. (2007). *Post-Combustion Evolution of Soot Properties in an Aircraft Engine*. https://api.semanticscholar.org/CorpusID:136958147

FlightGlobal.com. (2021). *Commercial engines*.

Gao, K., & Kanji, Z. A. (2022). Impacts of Cloud-Processing on Ice Nucleation of Soot Particles Internally Mixed With Sulfate and Organics. *Journal of Geophysical Research: Atmospheres*, *127*(22), e2022JD037146. https://doi.org/https://doi.org/10.1029/2022JD037146

Jing, L., El-Houjeiri, H. M., Monfort, J.-C., Littlefield, J., Al-Qahtani, A., Dixit, Y., Speth, R. L., Brandt, A. R., Masnadi, M. S., MacLean, H. L., Peltier, W., Gordon, D., & Bergerson, J. A. (2022). Understanding variability in petroleum jet fuel life cycle greenhouse gas emissions to inform aviation decarbonization. *Nature Communications*, *13*(1), 7853. https://doi.org/10.1038/s41467-022-35392-1

Kärcher, B. (1998). Physicochemistry of aircraft-generated liquid aerosols, soot, and ice particles: 1. Model description. *Journal of Geophysical Research: Atmospheres*, *103*(D14), 17111–17128. https://doi.org/https://doi.org/10.1029/98JD01044

Kärcher, B., Möhler, O., DeMott, P. J., Pechtl, S., & Yu, F. (2007). Insights into the role of soot aerosols in cirrus cloud formation. *Atmospheric Chemistry and Physics*, *7*(16), 4203–4227. https://doi.org/10.5194/acp-7-4203-2007

Mahrt, F., Kilchhofer, K., Marcolli, C., Grönquist, P., David, R. O., Rösch, M., Lohmann, U., & Kanji, Z. A. (2020). The Impact of Cloud Processing on the Ice Nucleation Abilities of Soot Particles at Cirrus Temperatures. *Journal of Geophysical Research: Atmospheres*, *125*(3), e2019JD030922. https://doi.org/https://doi.org/10.1029/2019JD030922

Petzold, A., Döpelheuer, A., Brock, C. A., & Schröder F. (1999). In situ observations and model calculations of black carbon emission by aircraft at cruise altitude. *Journal of Geophysical Research: Atmospheres*, *104*(D18), 22171–22181. https://doi.org/https://doi.org/10.1029/1999JD900460

Petzold, A., Ström, J., Ohlsson, S., & Schröder, F. P. (1998). Elemental composition and morphology of ice-crystal residual particles in cirrus clouds and contrails. *Atmospheric Research*, *49*(1), 21–34. https://doi.org/https://doi.org/10.1016/S0169-8095(97)00083-5

Pires, A. P. P., Han, Y., Kramlich, J., & Garcia-Perez, M. (2018). Chemical Composition and Fuel Properties of Alternative Jet Fuels. *Bioresources*, *13*(2), 2632–2657. https://doi.org/10.15376/biores.13.2.2632-2657

Testa, B., Durdina, L., Alpert, P. A., Mahrt, F., Dreimol, C. H., Edebeli, J., Spirig, C., Decker, Z. C. J., Anet, J., & Kanji, Z. A. (2024). Soot aerosols from commercial aviation engines are poor ice-nucleating particles at cirrus cloud temperatures. *Atmospheric Chemistry and Physics*, *24*(7), 4537–4567. https://doi.org/10.5194/acp-24-4537-2024

Virtanen, A., Joutsensaari, J., Koop, T., Kannosto, J., Yli-Pirilä, P., Leskinen, J., Mäkelä, J. M., Holopainen, J. K., Pöschl, U., Kulmala, M., Worsnop, D. R., & Laaksonen, A. (2010). An amorphous solid state of biogenic secondary organic aerosol particles. *Nature*, *467*(7317), 824–827. https://doi.org/10.1038/nature09455

Wong, H.-W., Yelvington, P. E., Timko, M. T., Onasch, T. B., Miake-Lye, R. C., Zhang, J., & Waitz, I. A. (2008). Microphysical Modeling of Ground-Level Aircraft-Emitted Aerosol Formation: Roles of Sulfur-Containing Species. *Journal of Propulsion and Power*, *24*(3), 590–602. https://doi.org/10.2514/1.32293

Yu, F., Turco, R. P., & Kärcher, B. (1999). The possible role of organics in the formation and evolution of ultrafine aircraft particles. *Journal of Geophysical Research: Atmospheres*, *104*(D4), 4079–4087. https://doi.org/https://doi.org/10.1029/1998JD200062